JGP Journal of General Physiology

# Distinct properties and activation of hexameric and heptameric Pannexin 1 channel concatemers

Smriti Gupta[1], Yu-Hsin Chiu[2], Mohan C. Manjegowda[1], Bimal N. Desai[1], Kodi S. Ravichandran[3], and Douglas A. Bayliss[1]

Pannexin 1 (PANX1) is a member of a topologically related and stoichiometrically diverse family of large pore membrane ion channels that support the flux of signaling metabolites (e.g., ATP) and fluorescent dyes. High-resolution structural analyses have identified PANX1 as a heptamer despite early evidence suggesting that it might be a hexamer. To determine if PANX1 channel activity is supported in both hexameric and heptameric conformations, we examined properties of concatenated PANX1 constructs comprising either six or seven subunits with intact or truncated C-termini (the latter to mimic caspase-cleavage activation). In whole-cell recordings from PANX1-deleted cells, the C-tail-truncated hexameric and heptameric concatemers generated outwardly rectifying PANX1-like currents only after severing the intersubunit linkers. Surprisingly, α1D adrenoceptor stimulation activated constructs with intact or truncated C-tails, even without linker cleavage. In inside-out patches from PANX1-deleted cells, linker cleavage activated C-tail truncated channels derived from either hexameric or heptameric concatemers. The heptamers presented peak unitary conductance and mean open time that was similar to channels assembled from the expression of unlinked single PANX1 subunits and greater than from the hexamers. In addition, the linker-cleaved heptameric concatemers supported greater PANX1-dependent ATP release and TO-PRO-3 uptake than the corresponding hexamers. These data indicate that functional PANX1 channels can be obtained in either hexameric or heptameric conformations and suggest that the distinct unitary properties of heptameric channels are more conducive to large molecule permeation by PANX1; they also suggest that there are distinct structural requirements for C-tail cleavage and receptor-mediated PANX1 activation mechanisms.

## Introduction

Pannexin 1 (PANX1) is a large pore membrane ion channel that, when activated, can release various metabolites for intercellular signaling in a multitude of (patho)physiological contexts (Koval et al., 2021; Laird and Penuela, 2021; Sanchez-Arias et al., 2021). For example, during cell apoptosis, caspase-mediated cleavage of channel subunits at a C-terminal site underlies an irreversible quantized activation process that supports release of ATP to attract local phagocytes for corpse clearance and release of various other molecules that support an anti-inflammatory environment (Chekeni et al., 2010; Qu et al., 2011; Yang et al., 2015; Chiu et al., 2017; Medina et al., 2020). PANX1 activation under these conditions allows for the uptake of fluorescent dyes that are used experimentally to mark apoptotic cells (Chekeni et al., 2010). In addition, in live cells, reversible activation mechanisms of intact PANX1 channels have been described, including by elevated [K⁺]e or [Ca²⁺]i, membrane stretch, or downstream of ionotropic and metabotropic receptors (reviewed in Chiu et al. [2018]; Dahl [2018]; Mim et al. [2021]; Wu et al. [2023]). In the resistance vasculature, the ATP release following α1D adrenoceptor (α1DAR)-mediated PANX1 activation is required for

sympathetic vasoconstriction and maintenance of arterial pressure (Billaud et al., 2015; Good et al., 2018); PANX1 activation by G protein-coupled receptors (GPCRs) is independent of C-tail cleavage and multiple intracellular signaling mechanisms and/or posttranslational modifications have been implicated (Billaud et al., 2015; Chiu et al., 2021; Medina et al., 2021).

PANX1 is a member of a broader group of large pore ion channels with four transmembrane helices that support the permeation of large molecules. Along with the other pannexin subunits (PANX2, PANX3), the extended family includes the connexins and invertebrate innexins that form gap junctions, the calcium homeostasis modulator (CALHM) channels, and volume regulatory SWELL1 (LRRC8) channels (reviewed in Syrjanen et al. [2021]). High-resolution structural analyses of this superfamily have revealed general similarities in the respective subunit membrane topology, with the notable exception of the CALHM channels; this structural work has also uncovered a remarkable diversity of channel stoichiometries, ranging from hexamers to undecamers (Syrjanen et al., 2021). The recognition that multiple stoichiometries are supported by

[1]Department of Pharmacology, University of Virginia, School of Medicine, Charlottesville, VA, USA;   [2]Institute of Biotechnology, National Tsing Hua University, Hsinchu, Taiwan;   [3]Division of Immunobiology, Department of Pathology and Immunology, Washington University, School of Medicine, St Louis, MO, USA.

Correspondence to Douglas A. Bayliss: dab3y@virginia.edu.

these topologically related channel subfamilies suggests the possibility that an individual channel type might itself be able to adopt multiple stoichiometries. In this respect, numerous high-resolution structural studies have consistently demonstrated that the pannexins form as heptamers (Deng et al., 2020; Jin et al., 2020; Michalski et al., 2020; Mou et al., 2020; Qu et al., 2020; Ruan et al., 2020), despite early low-resolution EM images, as well as indirect biochemical and fluorometric analyses, suggesting PANX1 subunits assemble into hexameric channels (Boassa et al., 2007; Wang et al., 2014; Chiu et al., 2017). Moreover, whole-cell currents and single-channel properties obtained from concatenated hexameric PANX1 constructs matched those from channels assembled from expression of single PANX1 subunits, consistent with the possibility that functional hexameric channels could be observed (Chiu et al., 2017).

Our previous work with hexameric concatemers of human PANX1 (hPANX1) subunits showed that those constructs could be successfully expressed as functional channels in HEK293T cells (Chiu et al., 2017). However, it is possible that endogenous hPANX1 subunit(s) in those cells were somehow incorporated into the holo-channel to achieve an alternative stoichiometry (e.g., heptameric). Thus, we tested whether hexameric concatemers can yield functional channels when expressed in a HEK293T cell line from which the hPANX1 gene was deleted by CRISPR/Cas9, and we compared the properties of those channels to heptameric concatemers. Our data reinforce the conclusion that PANX1 is functional in either hexameric or heptameric conformations; inasmuch as the intersubunit linkers in the concatemers provide a structural constraint on channel activation by C-tail removal, but not by α1DAR, they suggest that the channels can adopt structurally distinct gating conformations. Finally, our data reveal that the unitary channel properties and large molecule permeation capabilities of the heptamer most closely resemble channels derived from the expression of the single subunit, with larger conductance, longer mean open times, and more effective large molecule permeation than hexamers.

## Materials and methods

### Reagents
Chemicals used in this study were sourced from Sigma-Aldrich, unless otherwise specified. Other reagents were obtained as follows: polyethylenimine (PEI 25,000) from Polyscience (Cat#26008); Q-VD-OPh hydrate from Millipore Sigma (Cat#SML0063); Trovan mesylate provided by MedChem Express (Cat#HY-103399); Annexin V-PB (Cat#A35122) and Annexin binding buffer (Cat#V13246) from Thermo Fisher Scientific; Ectonucleotidase inhibitor ARL67156 from Millipore Sigma (Cat#A265); TO-PRO-3 Iodide (Cat#T3605) from Invitrogen; puromycin dihydrochloride (Cat#A1113803) and Zeocin (Cat#R25001) from Thermo Fisher Scientific; Ghost dye violet 450 from Cytek (Cat#13083-T100); Poly-L-Lysine hydrobromide from Millipore Sigma (Cat#P4707), CellTiter-Glo Luminescent Cell Viability Assay kit from Promega (Cat#G7570); and TEV protease and activated caspase 3 were provided by Dr. Susan Leonhardt (University of

Miami, Coral Gables, FL, USA), essentially as described (Chiu et al., 2017).

### Plasmids and cloning
Flag-tagged versions of full-length PANX1 and C-terminally truncated PANX1 were modified (C-terminally appended with eGFP), subcloned, and concatenated into pcDNA3 plasmids (Invitrogen) encoding either pentameric or hexameric PANX1 concatemers, as previously described (Chiu et al., 2017), to generate either hexameric or heptameric concatenated PANX1, respectively. Two versions of each of the concatenated pannexin constructs (hexamers and heptamers) were utilized in this study: one in which all subunit C-termini are intact, 6(6CT) and 7(7CT), and another in which all subunit all C-termini truncated at the caspase cleavage site, 6(0CT) and 7(0CT). The number in parentheses reflects the number of intact C termini in both hexamers and heptamers. The linkers joining adjacent subunits each contain a Flag epitope sequence (DYKDDDDK) as well as a TEV protease site (ENLYFQG); the linker joining the final subunit to a GFP tag substitutes a Tobacco Vein Mottling Virus (TVMV) protease site (ETVRFQS) to avoid removing the GFP reporter when TEVp is used to cut the linkers (see Fig. S1 for schematic representations of the concatemer constructs).

The pLX-sgRNA-BfuAI-2k plasmid (#112915; Addgene, provided by Dr. Ren-Jang Lin [Kurata and Lin, 2018]) was modified to encode two distinct small guide RNAs (sgRNAs) targeting the human *PANX1* gene. Initially, the blasticidin resistance gene in pLX-sgRNA-BfuAI-2k was replaced with a zeocin resistance marker, and an additional U6 promoter was subcloned into the plasmid, resulting in pLX-2sgRNA. Complementary oligonucleotides encoding the two sgRNAs (sgRNA 1: 5′-GCCTTCACC CAGTCACCGGC-3′; sgRNA 2: 5′-GATGGTCACGTGCATTGCGGT-3′) were then ligated into pLX-2sgRNA using T4 DNA ligase at the BsbI and BsmMI restriction sites as described (Chiu et al., 2021).

We have previously described the derivation of hPANX1-(TEV)-EGFP-FLAG-pEBB, in which the caspase cleavage sequence (IKMDVVD) was substituted with a TEVp cleavage site (ENLYFQG). The Red Lantern plasmid was prepared by substituting mCherry into pcDNA3.1-mGreenLantern and was a gift from Dr. Edward Perez-Reyes (University of Virginia, Charlottesville, VA). A codon-optimized TEVp (GenBank accession #DQ516974) in a modified pCI expression vector was a gift from Dr. Stephen Ikeda (Williams et al., 2009). A plasmid containing mouse α1DAR was obtained from OriGene (MR222643), as previously reported (Billaud et al., 2015; Good et al., 2018). All constructs were confirmed by DNA sequencing.

### Deletion of PANX1 from Cas9 stably expressed HEK293T cell line
HEK293T cells stably expressing CRISPR/Cas9 nuclease (co-expressing copGFP and puromycin resistant gene and integrated at the human AAVS1 Safe Harbor Locus) were obtained from GeneCopoeia (Cat# SL502). Cas9 HEK293T cells were transfected with the pLX-2sgRNA (containing two guide RNAs targeting hPANX1 as described earlier), and stable clones were selected using serial dilution method in media containing

puromycin (1 µg/ml) + zeocin (200–400 µg/ml). Deletion of PANX1 in Cas9 HEK293T cells was confirmed by western blot (Hsueh et al., 2024, *Preprint*).

## Cell culture and transfections

PANX1-deleted HEK293T cells and parental Cas9/GFP-HEK293T were cultured at 37°C, with humidified air containing 5% $CO_2$, in Dulbecco's modified Eagle medium (DMEM; Gibco) containing 10% fetal bovine serum (FBS; Gibco), penicillin, streptomycin, and sodium pyruvate (Gibco). HEK293T cells stably expressing Cas9 were maintained in selection media containing 1 µg/ml puromycin, while PANX1-deleted HEK293T cells were cultured in selection media with 1 µg/ml puromycin and 200–400 µg/ml Zeocin. Cells were transfected using PEI at 1:3 ratio (µg DNA: µL PEI) and studied 16–18 h after transfection. Cells were incubated with Q-VD-OPh (Q-VD, 20 µM) during and after transfection to prevent inadvertent C-terminal cleavage of PANX1 by endogenously activated caspases in all experiments; where noted, cells were also incubated during and after transfection with Trovan to block PANX1 channel activity.

## Electrophysiology

Transfected Cas9/GFP or hPANX1-deleted HEK293T cells were plated onto poly-L-lysine–coated coverslips that were placed in a chamber mounted on a Zeiss Axioskop with fluorescence and DIC optics; transfected hPANX1-deleted HEK293T or Cas9/GFP parental control cells were selected for recording based on GFP and Red Lantern fluorescence. All voltage-clamp recordings were conducted at room temperature using an Axopatch 200B amplifier, controlled by pCLAMP10 software, and digitized with a Digidata 1322A digitizer (all from Molecular Devices). Micropipettes were pulled from thin-walled, fire-polished borosilicate glass capillaries (Harvard Apparatus) using a P-97 Flaming/Brown Micropipette Puller (Sutter Instrument) and coated with Sylgard 184 silicone elastomer (Dow Corning Corporation).

Whole-cell recordings were performed using 3–5 MΩ borosilicate glass patch pipettes filled with an internal solution consisting of 100 mM CsMeSO$_4$, 30 mM TEA-Cl, 4 mM NaCl, 1 mM MgCl$_2$, 0.5 mM CaCl$_2$, 10 mM HEPES, 10 mM EGTA, 3 mM ATP-Mg, and 0.3 mM GTP-Tris (pH 7.3; ~290 mOsm). HEPES-based bath solution was composed of 140 mM NaCl, 3 mM KCl, 2 mM MgCl$_2$, 2 mM CaCl$_2$, 10 mM HEPES, and 10 mM glucose (pH 7.3). To activate PANX1 currents in α1DAR-expressing cells, phenylephrine (PE, 10 µM) was added to the bath solution; to block currents from transfected PANX1 channels, carbenoxolone (CBX) (100 µM) was added to the bath solution. Ramp voltage-clamp protocols were applied from –60 mV (0.4 mV/ms) at 7-s intervals. The CBX-sensitive current was defined as the difference in current at +80 mV before and after CBX application, normalized to cell capacitance (i.e., current density, pA/pF). All potentials were corrected for ~10 mV junction potential.

Inside-out patch recordings were obtained using micropipettes with a resistance of 7–10 MΩ. Data were filtered to 1 kHz using an eight-pole low-pass Bessel filter (LPF-8, Warner Instruments) and digitized at a sampling rate of 20 kHz. Pipettes were filled with the HEPES-based bath solution. After seal formation (≥10 GΩ), the membrane patch was excised and the bath solution was exchanged for an inside-out recording solution containing 150 mM CsCl, 5 mM EGTA, 10 mM HEPES, and 1 mM MgCl$_2$ (pH 7.3). Patches were held at negative and positive potentials between (from 20 to 80 mV), and TEVp (2 µg/ml) was applied near the patch under stop-flow conditions. Rare instances of channel activity in excised patches from concatemer-expressing cells observed before adding TEVp were considered non-specific; in addition, patches that displayed a lack of CBX sensitivity were excluded from the study. Data were analyzed using pCLAMP (v11.2) software.

## ATP release

PANX1-deleted HEK293T cells were transiently transfected with human PANX1-(TEV)-Flag construct or hPANX1 concatemers (with or without co-transfected TEVp). During transfection, cells were incubated with Q-VD (25 µM), to prevent cleavage of intact PANX1 C-termini, and Trovan (25 µM), to prevent PANX1 activity and depletion of cytosolic ATP. After transfection, cells were washed with assay buffer (1% bovine serum albumin in DMEM) containing either DMSO or Trovan. Following the washes, cells were incubated in an assay buffer containing the ectonucleotidase inhibitor, ARL67156 (300 µM), and either DMSO or Trovan for 4 h. Supernatants were collected for ATP measurement using the CellTiter-Glo Luminescent Cell Viability Assay (Promega) following the manufacturer's instructions. Trovan-sensitive (i.e., PANX1-dependent) ATP release was determined by subtracting the ATP concentration in cells treated with Trovan from that in cells treated with DMSO.

## Dye uptake

PANX1-deleted HEK293T cells were transiently transfected with human PANX1-(TEV)-Flag construct or hPANX1 concatemers (with or without co-transfected TEVp). Co-transfection was performed using a 10:1 ratio with Red Lantern. During transfection, cells were incubated with Q-VD (25 µM), to prevent cleavage of intact PANX1 C-termini, and with Trovan (25 µM), to prevent PANX1 activity. At 16–18 h after transfection, cells were subjected to a live–dead assay by staining with Ghost Violet dye on ice for 30 min. Post-staining, the cells were washed three times gently with an annexin-binding buffer. Subsequently, a dye cocktail containing Annexin V-Pacific Blue (Annexin V-PB) and TO-PRO-3 was applied to the cells at room temperature for 30 min before transferring them to ice. TO-PRO-3 uptake analysis was performed using the Cytek Aurora Full Spectral five-laser cytometer. Flow.Jo software (v.8) was used for gating and data analysis. The Ghost dye-positive cell population (representative of dead cells), Annexin V$^+$ cells (indicative of apoptotic cells), and low forward scatter (FSC) cells (indicative of cell debris) were excluded from analysis, while Ghost dye$^-$/Annexin-V$^-$/GFP$^+$/RL$^+$ cells (indicative of live PANX1-deleted HEK293T cells expressing PANX1 channel constructs) were analyzed for TO-PRO-3 dye uptake.

The experiment was performed in two sets to evaluate Trovan-sensitive dye uptake. In the first set, Trovan was included during both the transfection and staining processes. In the second set, Trovan was included during transfections but replaced with DMSO during the washes and dye staining steps.

Trovan-sensitive dye uptake was quantified by subtracting the mean fluorescence intensity (MFI) of TO-PRO-3 in Trovan-treated cells from that in DMSO-treated cells.

### Data analysis

Data are presented as mean ± SEM. Statistical analyses, described in figure legends, were performed using GraphPad Prism (v.10.3).

### Online supplemental material

Fig. S1 provides schematics of the concatemers and linker design. Fig. S2 shows that hexameric and heptameric concatemers generate comparable whole-cell currents in control Cas9/GFP cells. Fig. S3 illustrates single channel properties of hPANX1(-TEV) transfected as a single subunit in hPANX1-deleted HEK293T cells.

## Results

### Whole-cell currents from C-truncated hexameric and heptameric PANX1 constructs are indistinguishable in PANX1-deleted HEK293T cells

As previously described (Chiu et al., 2017), the hexameric concatemer constructs comprise six hPANX1 subunits that are all either full-length or truncated at the C-terminal caspase site (termed 6(6CT) and 6(0CT), respectively); the subunits are joined in a tail-to-head orientation via a linker that contains a Flag epitope tag and a Tobacco Etch Virus protease (TEVp) cleavage site (Fig. S1). We transfected 6(6CT) or 6(0CT) constructs in either parental Cas9/GFP-expressing HEK293T cells or in the corresponding hPANX1-deleted HEK293T cells, and the cells were incubated with Q-VD-OPh (20 μM) following transfection to interfere with inadvertent caspase-mediated cleavage of intact C-tails (e.g., in the 6(6CT) channels). We performed whole-cell recordings with TEVp added to the pipette solution to cut the cytoplasmic linkers between adjacent subunits in channels present on the cell membrane (Fig. 1 A and Fig. S2 A). In a previous work, we found that severing the intersubunit linker was necessary for the activation of concatenated channels; this was true even for constructs in which all constituent subunits are truncated at the caspase cleavage site (e.g., 6(0CT)), suggesting that linking the N- and C-terminal domains in these concatemers added a structural constraint to channel activation (Chiu et al., 2017).

In whole-cell recordings from parental Cas9/GFP cells expressing the 6(0CT) constructs, a CBX-sensitive current with membrane current–voltage (I-V) characteristics typical of PANX1 channels developed over time as TEVp dialyzed into the cell (Fig. S2 A). As expected, this time-dependent current was not observed in 6(0CT)-expressing cells recorded in the absence of TEVp, or in 6(6CT)-expressing cells, recorded either with or without TEVp (Fig. S2 A and Fig. 1 C). These data are consistent with earlier observations suggesting that the intersubunit linkers provide a structural constraint that must be removed for activation of C-tail truncated concatemeric hPANX1 channels. Moreover, they also indicate that linker cleavage cannot activate 6(6CT) channels with their fully intact C-tails.

We conducted similar experiments with 6(6CT) and 6(0CT) constructs in hPANX1-deleted HEK293T cells, which yielded an essentially identical pattern of outcomes (Fig. 1, A and C). We observed hPANX1-like CBX-sensitive currents only from cells expressing the 6(0CT) constructs and only when the pipette contained TEVp to cut the intersubunit linkers. We also noted that the hPANX1 current density (pA/pF, corrected for cell size) was lower in the hPANX1-deleted cells in comparison to that seen in parental Cas9/GFP cells (by ~30%; 61.2 ± 5.9 pA/pF versus 87.3 ± 8.3 pA/pF, $n$ = 9 & 13) (Fig. 1 C).

For comparison, we prepared corresponding heptameric 7(7CT) and 7(0CT) concatemers and recorded TEVp-activated currents in Cas9/GFP and hPANX1-deleted HEK293T cells following procedures identical to those described above (Fig. S2 B and Fig. 1 B). The results with these constructs were qualitatively identical to those obtained with 6(6CT) and 6(0CT) constructs. That is, the 7(0CT) concatemer generated time-dependent, CBX-sensitive hPANX1-like current in both the Cas9/GFP and hPANX1-deleted cells when TEVp was included in the recording pipette; those currents were not observed with 7(0CT) in the absence of TEVp or with the 7(7CT) constructs under any conditions (Fig. S2 B; and Fig. 1, B and D). The current density was again lower in the hPANX1-deleted cells relative to that recorded in Cas9/GFP cells (by ~33%; 42.8 ± 6.0 pA/pF versus 74.8 ± 11.3 pA/pF, $n$ = 9 & 7) (Fig. 1 D).

Notably, the TEVp-activated current density was lower for both 6(0CT) and 7(0CT) constructs in hPANX1-deleted cells in comparison to Cas9/GFP parental cells (Fig. 1, C and D), and there was also no difference between 6(0CT) and 7(0CT) current density when compared in either cell line (Fig. 1 E). This suggests that the differences in current density between cell lines were not likely due to an endogenous subunit inserting itself specifically into the hexameric concatemer since this was also seen with the heptameric construct. Rather, we suspect that the reduced current density simply reflects the generally lower expression of all constructs in the hPANX1-deleted cells.

Overall, these data are consistent with previous whole-cell recordings of these hexameric concatemers in HEK293T cells (Chiu et al., 2017). Moreover, comparisons of TEVp-activated currents from either 6(0CT)-expressing or 7(0CT)-expressing between Cas9/GFP and hPANX1-deleted HEK293T cells revealed no obvious contribution from the endogenous hPANX1 subunit.

### TEVp-mediated linker cleavage is not required for GPCR activation of PANX1 concatemers

The 6(6CT) and 7(7CT) constructs were silent in the presence of TEVp. Although we expect that this was due to the presence of fully intact C-tails in these linker-cleaved concatemers, a trivial alternative possibility is that the constructs were not expressed or not properly trafficked to the plasma membrane. To rule this out, we invoked a second mechanism of PANX1 activation that can occur in channels with intact C-tails, specifically, PANX1 activation by GPCRs (Billaud et al., 2015; Chiu et al., 2017, 2021).

In whole-cell recordings with TEVp-containing pipettes from hPANX1-deleted cells that were co-expressing the α1DAR, we again saw no time-dependent development of currents from either the 6(6CT) and 7(7CT) concatemers. However, an

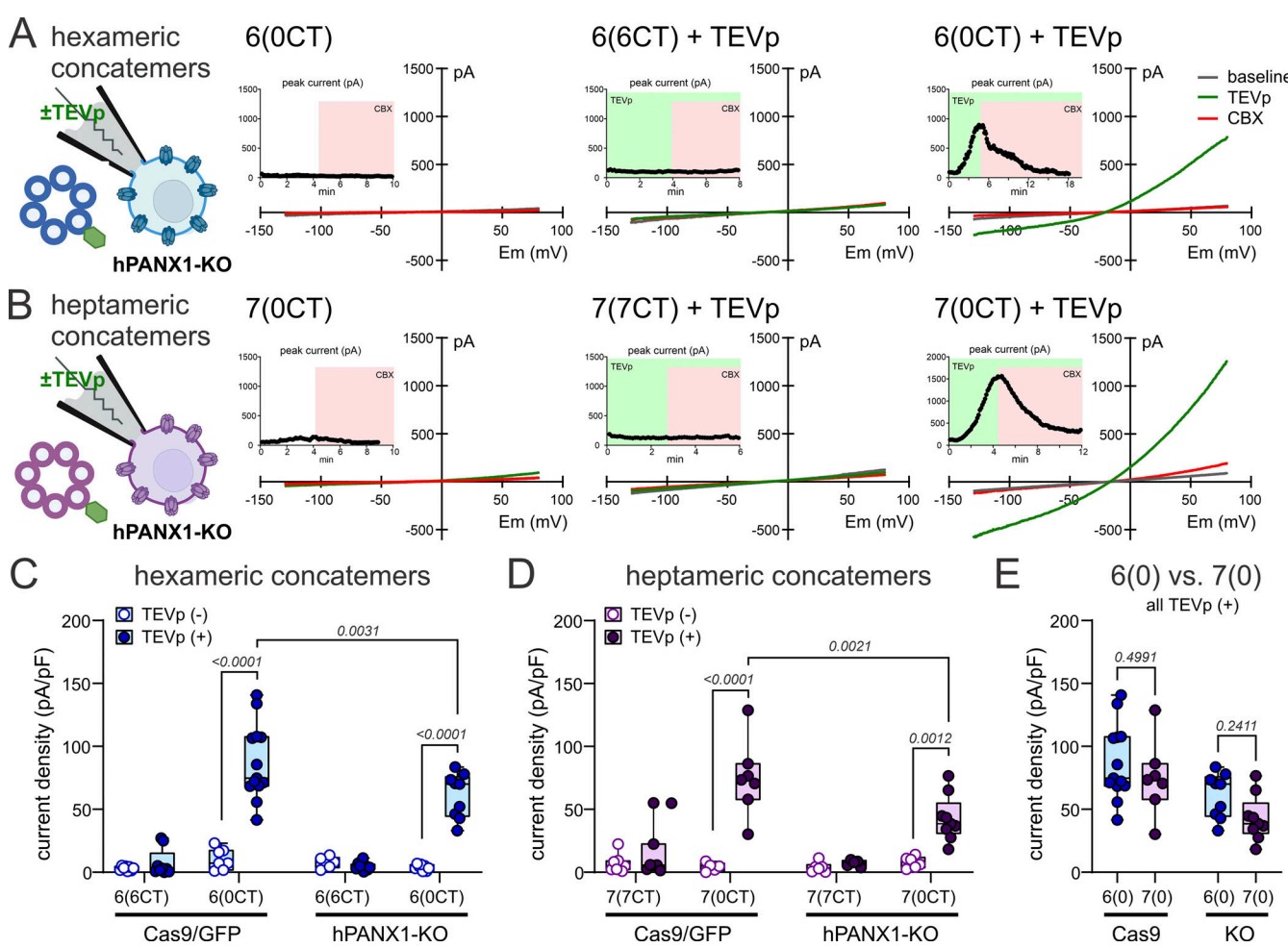

Figure 1. **Hexameric and heptameric concatemers generate comparable whole-cell currents in control and hPANX1-deleted HEK293T cells. (A and B)** Schematic of configuration for whole-cell recording of hPANX1-deleted HEK293T cells with TEVp in the pipette to cleave linkers of concatenated hexameric (A) and heptameric (B) hPANX1 channels; exemplar ramp *I*-V curves and time series of peak current at +80 mV (insets) from the indicated constructs and conditions. Note that CBX-sensitive hPANX1-like currents developed only in cells expressing the 6(0CT) and 7(0CT) constructs with truncated C-tails, and only when the pipette contained TEVp to cut the inter-subunit linkers. **(C and D)** Quantification of the peak CBX-sensitive current density (pA/pF) from hexameric (C) and heptameric (D) hPANX1 concatemers under the indicated conditions. Statistical analysis by three-way ANOVA with Šídák's multiple comparisons test. **(E)** Comparison of current density from hexameric and heptameric in either control (Cas9/GFP) or hPANX1-deleted HEK293T cells. Statistical analysis by two-way ANOVA with Šídák's multiple comparisons test.

hPANX1-like, CBX-sensitive current was observed after the addition of PE, an α1DAR agonist (Fig. 2, A–D). PE did not activate PANX1 current in concatemer-expressing cells that had not been transfected with α1DAR and, conversely, exposure to TEVp without additional PE did not evoke a current in α1DAR-expressing cells (Fig. 2, B and D). Unexpectedly, we found that this α1DAR-mediated activation could also occur for all intact and C-truncated hexameric and heptameric concatemers, even when TEVp was not included in the pipette (Fig. 2, A–D). This was surprising because the intersubunit linkers were not cleaved by TEVp in these recording conditions, as required for activation by C-tail deleted channels; this is illustrated by the absence of PANX1 channel activation prior to PE addition with either the 6(0CT) or 7(0CT) channels (Fig. 2, A–D). Thus, in addition to confirming expression and membrane trafficking of the C-tail intact concatemeric channels, these data also suggest that channel activation by GPCRs, unlike that by C-tail removal,

is not constrained by tethering the hPANX1 C-termini to the N-termini of adjacent subunits in linked concatemeric channels.

## Single-channel currents from heptameric PANX1 concatemers have larger conductance and longer mean open times than those from hexameric concatemers

The characteristics of whole-cell currents recorded from hexameric and heptameric concatemers were essentially indistinguishable in either Cas9/GFP or hPANX1-deleted HEK293T cells. However, we considered the possibility that there may be differences in the unitary properties of the underlying channels and made single-channel recordings from hPANX1-deleted HEK293T cells expressing either the 6(0CT) or 7(0CT) constructs. We again used knockout cells to avoid any contributions from endogenous hPANX1 subunits that might have gone undetected in whole-cell recordings. For comparison, we also recorded hPANX1 channels in knockout cells after transfecting the

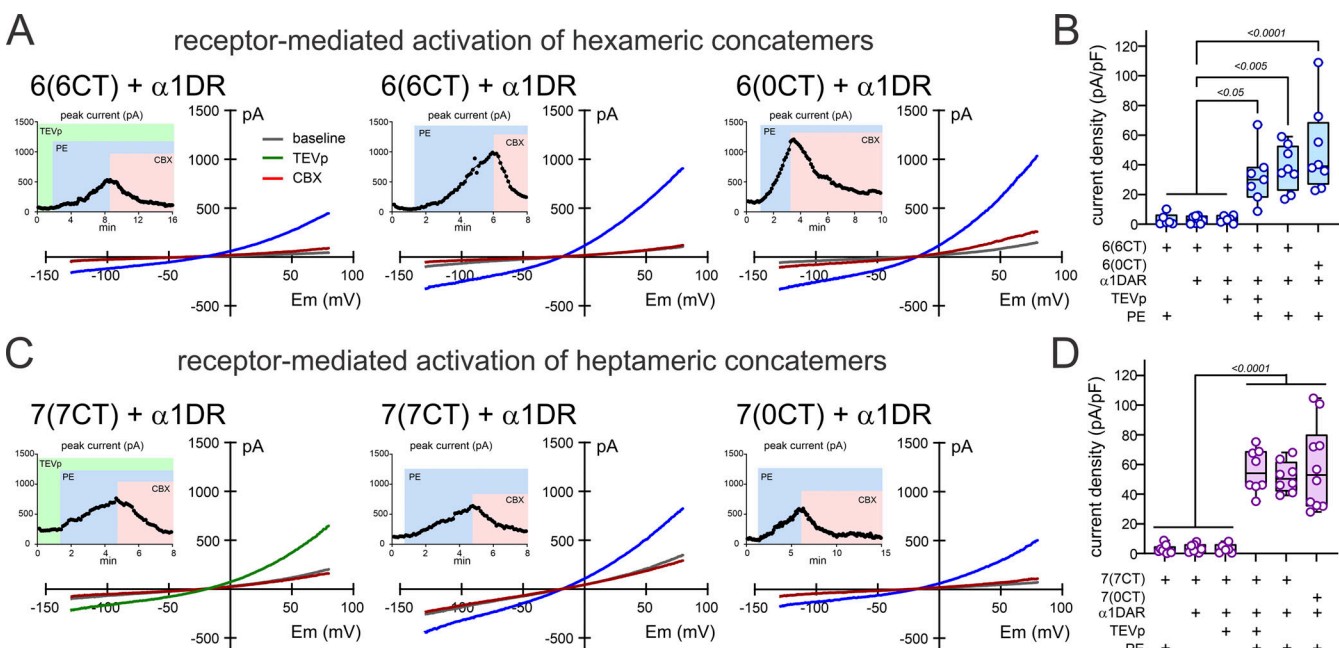

**Figure 2. In hPANX1-deleted HEK293T cells, receptor-activated currents are evoked in hexameric and heptameric concatemers with intact intersubunit linkers. (A and C)** Whole-cell recordings of PE-activated, CBX-sensitive currents in α1DAR-expressing hPANX1-deleted HEK293T cells cotransfected with concatenated hexameric (A) and heptameric (C) hPANX1 channels; exemplar ramp I-V curves and time series of peak current at +80 mV (insets). **(B and D)** Quantification of the peak CBX-sensitive current density (pA/pF) from hexameric (B) and heptameric (D) hPANX1 concatemers under the indicated conditions. Statistical analysis by ANOVA with Tukey's multiple comparisons test. Note that PANX1-like currents developed even when the intersubunit linkers were intact (i.e., when the pipette did not contain TEVp).

corresponding single hPANX1 subunit (i.e., not expressed as a concatemer).

The 6(0CT) channel was basally silent when excised in the inside-out configuration (at +80 mV) from hPANX1-deleted HEK293T cells, and CBX-sensitive channel activity became apparent when TEVp was introduced into the bath solution under stop-flow conditions to cut the intersubunit linkers (Fig. 3 A). The TEVp-dependence and CBX-sensitivity is in agreement with our whole-cell recordings of 6(0CT) and consistent with channel activity derived from the expressed hPANX1 concatemers; it is also similar to earlier single channel recordings obtained after expressing the hexameric 6(0CT) construct in parental HEK293T cells (Chiu et al., 2017).

We recorded 6(0CT) channel activity over a range of patch potentials (from –80 to +80 mV) to characterize unitary conductance, open probability ($P_O$), and mean open time (Fig. 3 B). Single-channel current amplitudes were determined from all points histograms (not shown) and plotted against patch potential (Fig. 3 C). The outward current amplitudes at positive potentials were larger than the inward current amplitudes at comparable negative potentials, resulting in single-channel current voltage (i-V) relationship with obvious outward rectification. Specifically, a linear fit through the outward currents yielded a conductance of 55.5 ± 0.5 pS, and that through the inward currents was 17.6 ± 0.6 pS. An outwardly rectifying i-V relationship with a peak conductance <100 pS was expected from our previous inside-out recordings of channels from single wild-type hPANX1 subunits or 6(0CT) constructs in HEK293T cells (Chiu et al., 2017). However, the 6(0CT) channels recorded

here had a smaller outward conductance (∼55 pS) than observed previously for either wild-type hPANX1 or 6(0CT) constructs in HEK293T cells (∼95 pS) (Chiu et al., 2017). We suspect that properties attributed to the hexameric channels following TEV cleavage in that earlier work may have arisen instead from heptameric channels formed after the rapid incorporation of endogenous hPANX1 monomers into the cleaved hexameric constructs.

The TEVp-activated 6(0CT) hPANX1 channels transitioned between closed and open states, and an events analysis showed no obvious voltage dependence of the channel $P_O$ across the recorded patch potentials (Fig. 3 D). From the open time distribution, we obtained comparable mean open times of ∼2 ms at +60 and +80 mV (range: 1.3–4.4 ms) that were longer than those seen at the corresponding negative potentials (∼0.5 ms, range: 0.4–0.6 ms) (Fig. 3, E and F). The shorter mean open times at negative potentials were countered by more frequent openings (not shown) to account for the preserved $P_O$ across membrane potentials.

We performed the same experiments and analysis with 7(0CT) concatemers in hPANX1-deleted HEK293T cells (Fig. 4) and found a general pattern of activity that was in many ways similar to that described above for 6(0CT) channels. That is, 7(0CT) channels were silent following excision in inside-out patches, activated by exposure to TEVp and inhibited by bath application of CBX (Fig. 4 A); the channels were outwardly rectifying, with peak conductance at depolarized potentials <100 pS (Fig. 4, B and C); and the $P_O$ was unaffected by patch potential (Fig. 4 D), with mean open times that were longer at depolarized

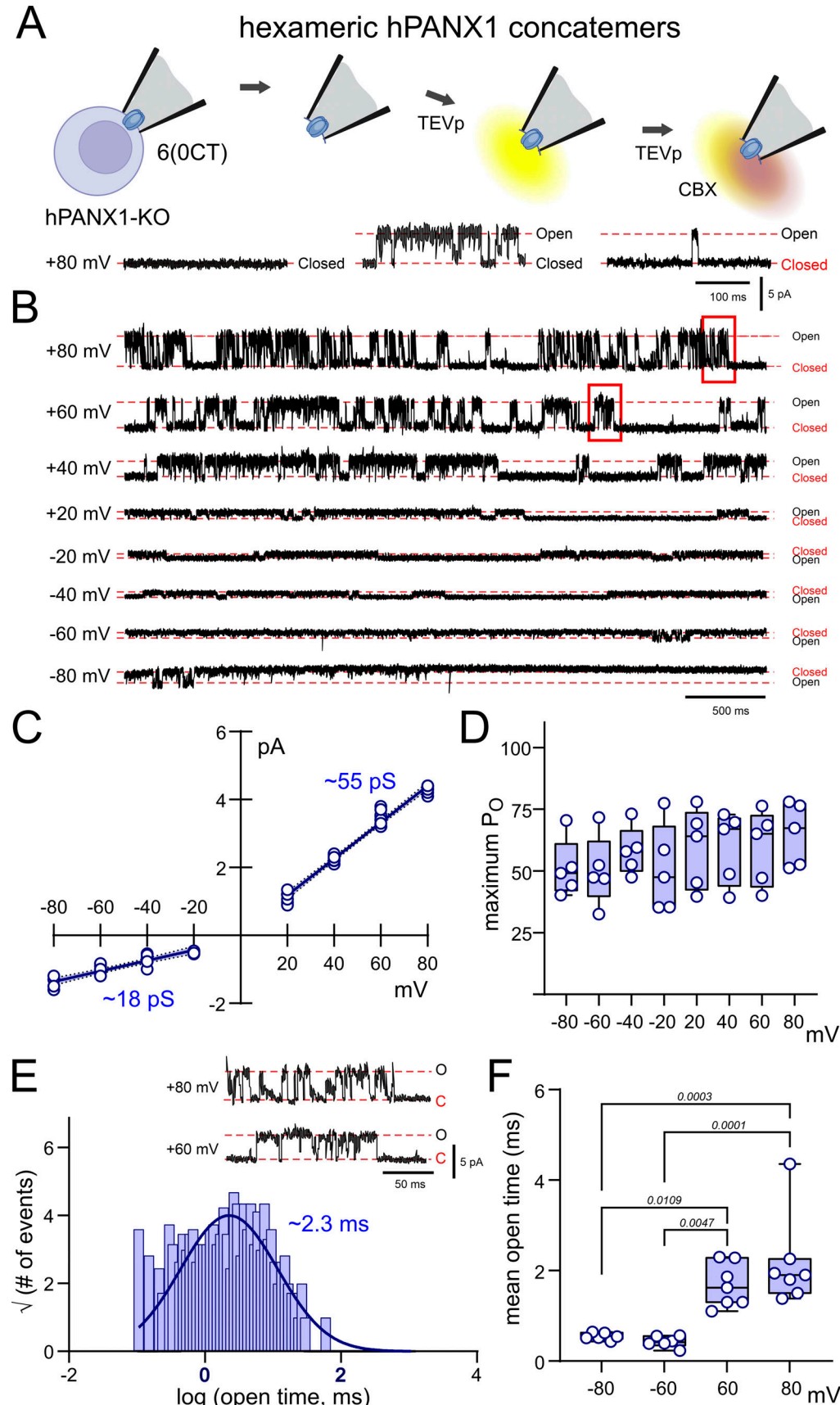

Figure 3. **Unitary channel properties of hexameric hPANX1 concatemers in hPANX1-deleted HEK293T cells. (A)** Inside-out patch recordings of TEVp-activated CBX-sensitive channels from hPANX1-deleted HEK293T cells expressing concatenated hexameric hPANX1 channels. **(B)** Recordings of

TEVp-activated channels at the indicated membrane potentials. **(C)** Single channel outwardly rectifying *i*-V relationship for hexameric hPANX1 concatemers, with estimated conductance from the fitted linear regression at positive and negative potentials (shaded regions is the 95% confidence interval). **(D)** Channel $P_O$ at the indicated membrane potentials. **(E)** Open time distribution fitted with a Gaussian function to estimate channel mean open time. Insets: Expanded channel records at a faster time base (from region indicated in B). **(F)** Channel mean open times at the indicated membrane potentials. Statistical analyses by ANOVA with Tukey's multiple comparisons test.

---

potentials (Fig. 4, E and F). Despite these qualitative similarities to the 6(0CT) channels, there were quantitative differences. For the 7(0CT) channels, the outwardly rectifying profile of the *i*-V relationship was less pronounced than for the 6(0CT), with a higher conductance at both depolarized (84.8 ± 1.6 pS) and hyperpolarized potentials (45.2 ± 1.3 pS) (Fig. 4 C). In addition, and most strikingly, the channel mean open times were substantially longer at both depolarized (∼15 ms, range: 12–20 ms) and hyperpolarized potentials (∼7–8 ms, range: 6–12 ms) (Fig. 4 F). Moreover, the peak conductance obtained from these 7(0CT) channels (∼86 pS) was closer to that recorded earlier from wild type hPANX1 in HEK293T cells (∼95 pS) (Chiu et al., 2017).

We also made inside-out recordings of TEVp-activated, CBX-sensitive channels from hPANX1-deleted HEK293T cells expressing hPANX1(TEV) as a single subunit (Fig. S3) for comparison with the properties of the 6(0CT) and 7(0CT) concatemeric channels. For TEVp-activated channels derived from single hPANX1(TEV) subunits, single-channel conductance was 73.7 ± 1.6 pS and 54.9 ± 2.2 pS at depolarized and hyperpolarized potentials, respectively (Fig. S3, A–C), the $P_O$ was voltage-independent (Fig. S3 D), and the channels displayed long mean open times (∼16 at +60 and +80 mV, range: 11–22 ms; ∼8 ms at −60 and −80 mV, range: 5–11 ms) (Fig. S3, E and F).

The data from all channel types recorded in hPANX1-deleted HEK293T cells are overlaid in Fig. 5. Clearly, the properties of channels derived from expressing the single hPANX1(TEV) subunit and from the 7(0CT) concatemer are most similar to each other, with both presenting higher conductance (Fig. 5, B and C) and longer mean open times (Fig. 5, D and E) than 6(0CT) concatemers at all membrane potentials examined. Notably, we did not observe any change in channel properties over time in these recordings, or when we added additional TEVp to the bath, suggesting that the differences in channel properties in 6(0CT) constructs are unlikely due to variable or incomplete linker cleavage by the protease.

Collectively, these data demonstrate that it is possible for a hexameric concatemer of hPANX1 to form a functional channel in hPANX1-deleted HEK293T cells. However, whereas the single channel properties derived from the 6(0CT) constructs are generally similar to those obtained from either the heptameric 7(0CT) concatemer or the non-linked single hPANX1 subunit, they are quantitatively distinct in presenting a smaller conductance and markedly shorter mean open times at all membrane potentials.

**Large molecules permeate via heptameric PANX1 channels more effectively than through the corresponding hexamers**

Pannexin channels are renowned for large molecule permeability. When activated by C-tail cleavage during apoptosis, they support the release of various metabolites that function as signaling molecules (e.g., adenosine triphosphate, ATP) and uptake of fluorescent dyes used experimentally to mark dying cells (e.g., TO-PRO-3) (Chekeni et al., 2010; Medina et al., 2020; Narahari et al., 2021). We examined ATP release and dye uptake in hPANX1-deleted HEK293T cells expressing either 6(0CT) or 7(0CT) hPANX1 concatemers as well as the single unlinked hPANX1(TEV) subunit. The cells were also co-transfected with an empty vector (pcDNA3) or a TEVp-expressing construct to cleave the intersubunit linkers (or the C-tails of channels formed from expressing the single subunit); they were again incubated in Q-VD to inhibit caspase activity and also incubated in Trovan (25 μM) after transfection to block permeation of atomic ions and signaling metabolites via the TEVp-activated hPANX1 channels (Poon et al., 2014). For ATP release and dye uptake experiments, cells either remained under Trovan-mediated channel block or Trovan was removed to allow hPANX1-dependent permeation (defined as the difference ± Trovan).

We performed luciferase assays to measure ATP release via hPANX1 constructs in transfected hPANX1-deleted HEK293T cells (Fig. 6 A). In cells that were co-transfected with TEVp, we found elevated ATP release from all constructs after replacing Trovan with vehicle (dimethyl sulfoxide, DMSO)-containing buffer (Fig. 6 A, left). In the continued presence of Trovan, ATP release was uniformly low and not different from the corresponding cells that had not been transfected with TEVp (Fig. 6 A, middle). The Trovan-sensitive, TEVp-activated ATP release (i.e., DMSO-Trovan; Fig. 6 A, right) was significantly lower in cells expressing 6(0CT) concatemers than that observed from cells expressing either the single hPANX1(TEV) subunit or the 7(0CT) concatemer.

We performed flow cytometry for analysis of TO-PRO-3 dye uptake from TEVp-activated hPANX1 concatemers (Fig. 6 B). In this case, cells were also co-transfected with Red Lantern, which was used as a positive marker of transfected cells; Annexin V was used as a negative marker to exclude any apoptotic cells. In the absence of Trovan, the TEVp-activated 6(0CT) and 7(0CT) concatemers took up significantly more dye than their counterparts that were not transfected with TEVp (Fig. 6 B, left), or which were maintained in the continued presence of Trovan (Fig. 6 B, right). Likewise, co-transfection of TEVp in cells expressing the single hPANX1(TEV) subunit was also associated with Trovan-sensitive dye uptake. Again, the Trovan-sensitive, TEVp-activated dye uptake (i.e., DMSO-Trovan; Fig. 6 B, right) was less for the 6(0CT) concatemer than for either the 7(0CT) concatemer or the single hPANX1(TEV) subunit.

Notably, the 7(0CT) concatemer displayed similar $P_O$ as the 6(0CT) concatemer, with larger single channel conductance (see Fig. 2, Fig. 3, and Fig. 4). However, whole-cell current density obtained with the 7(0CT) construct was not larger in amplitude (Fig. 1 E). This suggests that expression levels for the 7(0CT)

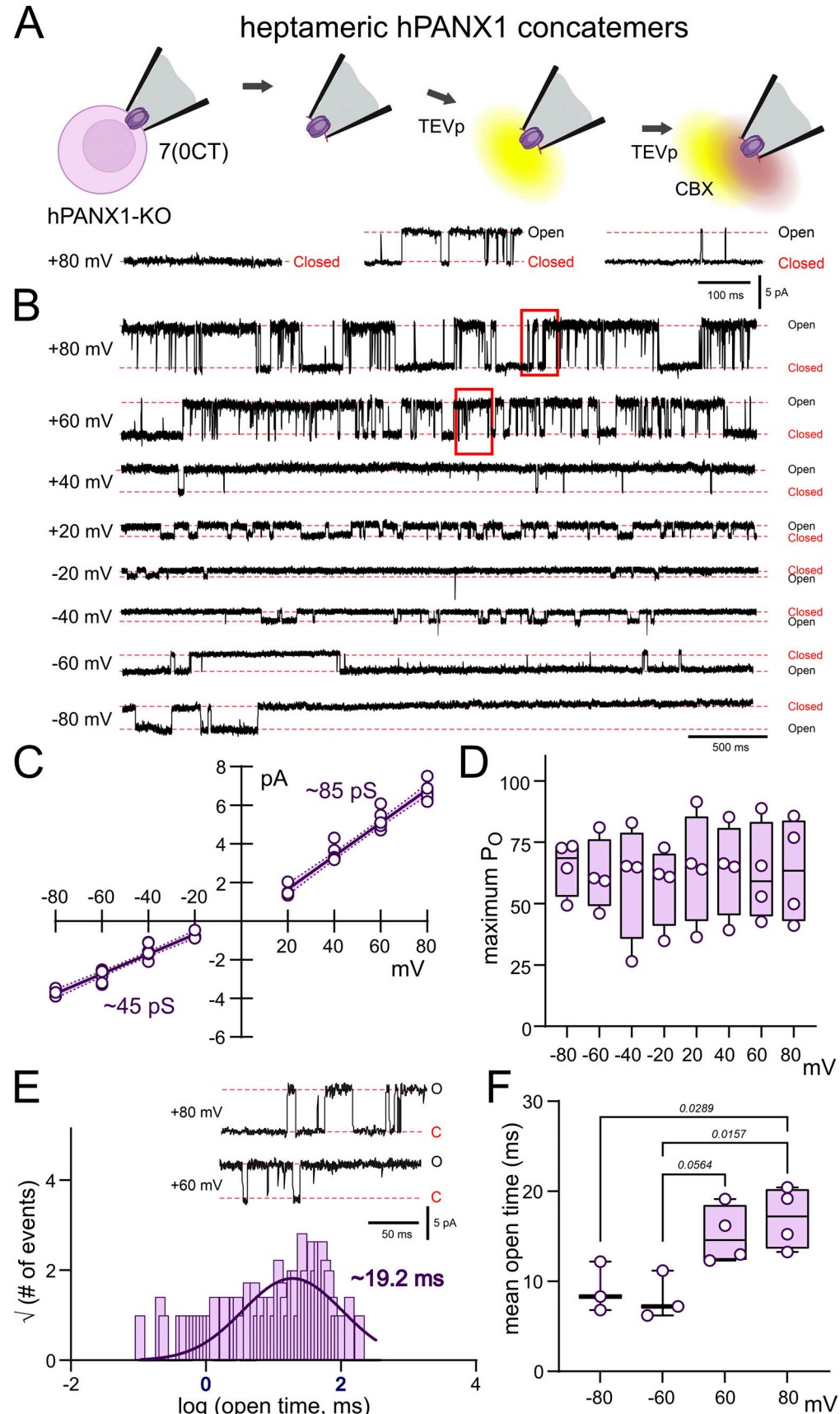

Figure 4. **Unitary channel properties of heptameric hPANX1 concatemers in hPANX1-deleted HEK293T cells. (A)** Inside-out patch recordings of TEVp-activated CBX-sensitive channels from hPANX1-deleted HEK293T cells expressing concatenated heptameric hPANX1 channels. **(B)** Recordings of

TEVp-activated channels at the indicated membrane potentials. **(C)** Single channel *i*-V relationship for heptameric hPANX1 concatemers, with estimated conductance from the fitted linear regression at positive and negative potentials (shaded regions is the 95% confidence interval). **(D)** Channel $P_O$ at the indicated membrane potentials. **(E)** Open time distribution fitted with a Gaussian function to estimate channel mean open time. Insets: Expanded channel records at a faster time base (from region indicated in B). **(F)** Channel mean open times at the indicated membrane potentials. Statistical analyses by ANOVA with Tukey's multiple comparisons test.

concatemer were lower, perhaps due to the increased size of the concatenated polypeptide. Thus, the enhanced large molecule permeation observed with the 7(0CT) concatemer, relative to 6(0CT), is not likely due to differences in expression, and indeed may even have been underestimated due to lower expression.

## Discussion

These experiments present evidence that functional PANX1 channels can be obtained from both hexameric and heptameric concatemer constructs, even when expressed in PANX1-deleted cells where there is no opportunity for an endogenous subunit to "infiltrate" the channel. The channels derived from the different PANX1 concatemers produced generally similar whole cell and unitary properties (weakly outwardly rectifying conductance, with peak <100 pS; Po independent of voltage). However, they were not identical. Most notably, the heptamer displayed larger unitary conductance at positive and negative membrane potentials, and ~10-fold longer mean open times. These distinctly different channel properties may be responsible for the ability of the heptameric concatemer to better support the large molecule permeation (ATP release, dye uptake) characteristic of PANX1. Our data thus suggest the possibility that native PANX1 channels may form as hexamers as well as heptamers, at least in some contexts, with potential implications for PANX1-dependent metabolite release and intercellular signaling.

In this respect, it is notable that recent observations have suggested alternative stoichiometries and channel properties for some other structurally well-defined channels. For example, a tetrameric conformation of the ligand-gated 5-HT3 receptor channel was observed in addition to its more typical pentameric conformation (Introini et al., 2024). Conversely, the typically tetrameric TRPV3 channel was found to also exist in a pentameric conformation, in this case via dynamic interchange by channel protomers within a membrane bilayer (Lansky et al., 2023). In addition, and outside of channels, the viral CA protein can exist in either hexameric or pentameric conformations to shape the HIV capsid (Pornillos et al., 2009). So, the presence of multiple stoichiometrically distinct protein conformations may be more common than currently recognized, including in ion channels (Lansky et al., 2023; Introini et al., 2024).

### Interpretive caveats and limitations

It is important to acknowledge some caveats with experiments based on concatemeric channel constructs. First, although our use of PANX1-deleted cells as a context for the concatemer expression obviates concern with contributions from endogenous subunits, it remains formally possible that the hexameric concatemers "borrowed" a subunit from a nearby concatemer to fulfill a heptameric requirement. For our recording experiments, it is unlikely that this could have happened during channel biogenesis since the channels remained linked (and inactive) in either the whole cell or the excised patch configuration until TEVp was dialyzed/applied to cut the linkers, with channel activity becoming apparent over seconds to minutes. However, some protomer exchange between the TEVp-freed subunits during the course of the experiment may be possible, especially in light of the recent observations that TRPV3 channels can transition between tetrameric and pentameric conformations in membranes as quickly as 2–3 min (Lansky et al., 2023). Even so, the consistent and distinct single-channel properties obtained, respectively, from expressing either the hexameric or heptameric concatemers are most consistent with recordings derived from faithful production of channels in the expected hexameric and heptameric conformations.

For our ATP release and dye uptake studies, the TEVp construct was co-transfected with the concatemers, and thus the protease was present from the time it was expressed until we performed the assays. Indeed, we observed large constitutive PANX1 currents in earlier experiments in which TEVp was co-transfected with PANX1(TEV) channels, indicating protease activity, C-tail cleavage, and channel activation at some point following transfection (Chiu et al., 2014, 2017). In the current experiments, we incubated cells in the PANX1 blocker Trovan after transfection to avoid the consequences of channel activation on ATP/metabolite release and cell health (Poon et al., 2014). It is possible, however, that TEVp-mediated linker cleavage liberated subunits from the concatemers to form channels in some preferred stoichiometric conformation. We think it is unlikely that this can account for the differences we observed in large molecule permeation since cells expressing hexameric concatemers showed significantly lower ATP release and dye uptake even though whole-cell current density was equivalent to those expressing the heptameric constructs. This suggests that the reduced ATP release and dye uptake may instead reflect different properties intrinsic to the hexamer (e.g., smaller conductance and shorter mean open times).

### Comparison with previous PANX1 channel descriptions

From inside-out patch recordings of hPANX1 channels activated by C-tail cleavage, our data have consistently revealed an outwardly rectifying unitary conductance that is <100 pS at positive membrane potentials when the current is outward, and even smaller at negative potentials when the current is inward (Chiu et al., 2014, 2017, 2018, 2021). This was again the case here for C-tail truncated channels that were derived from hexameric (~54 and ~16 pS) or heptameric (~86 and ~51 pS) concatemers, or from the expression of a single subunit (~72 and ~46 pS). In the present recordings, the outward conductance for all constructs

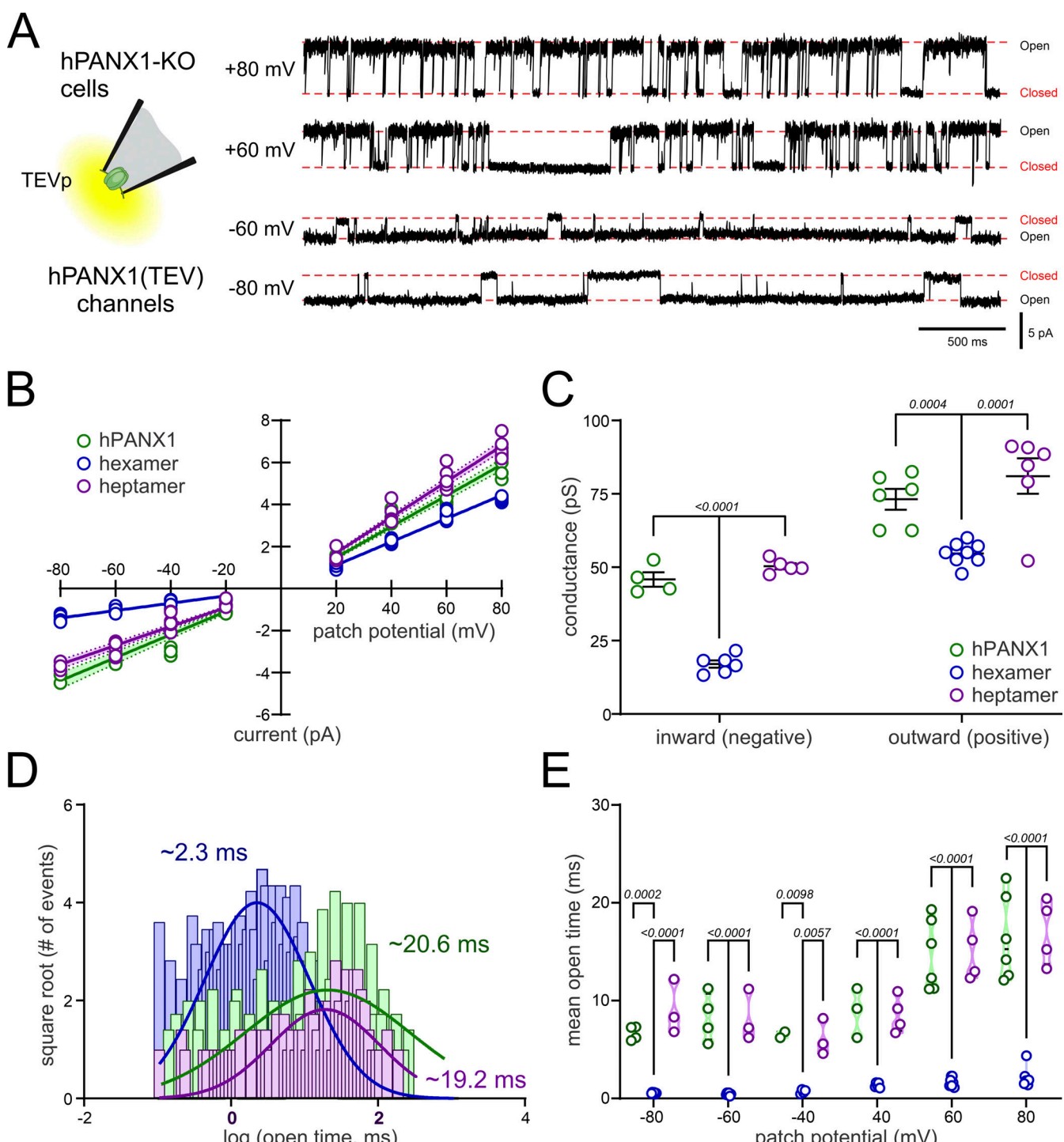

Figure 5. **Heptameric concatemers closely match unitary properties of hPANX1 channels in hPANX1-deleted HEK293T cells. (A)** Inside-out patch recordings of TEVp-activated channels from hPANX1-deleted HEK293T cells expressing hPANX1(TEV) channels at the indicated membrane potentials. **(B)** Overlay of *i*-V relationships for TEVp-activated hPANX1(TEV) channels and hexameric and heptameric hPANX1 concatemers, with linear regressions fitted at positive and negative potentials (shaded regions is the 95% confidence interval; data from Fig. 3, Fig. 4, and Fig. S2). **(C)** Unitary conductance estimates from individual patches expressing the indicated construct; values from hPANX1 and the heptamer were significantly greater than those from the hexamer. Analysis by two-way ANOVA with Šídák's multiple comparisons test. **(D)** Overlay of open time distributions from the indicated constructs recorded at +80 mV, with their corresponding Gaussian fits and channel mean open times. (Data from Fig. 3, Fig. 4, and Fig. S2). **(E)** Channel mean open times for hPANX1(TEV) and the hexameric and heptameric concatemers at the indicated potentials. Analysis by two-way ANOVA with Tukey's multiple comparisons test.

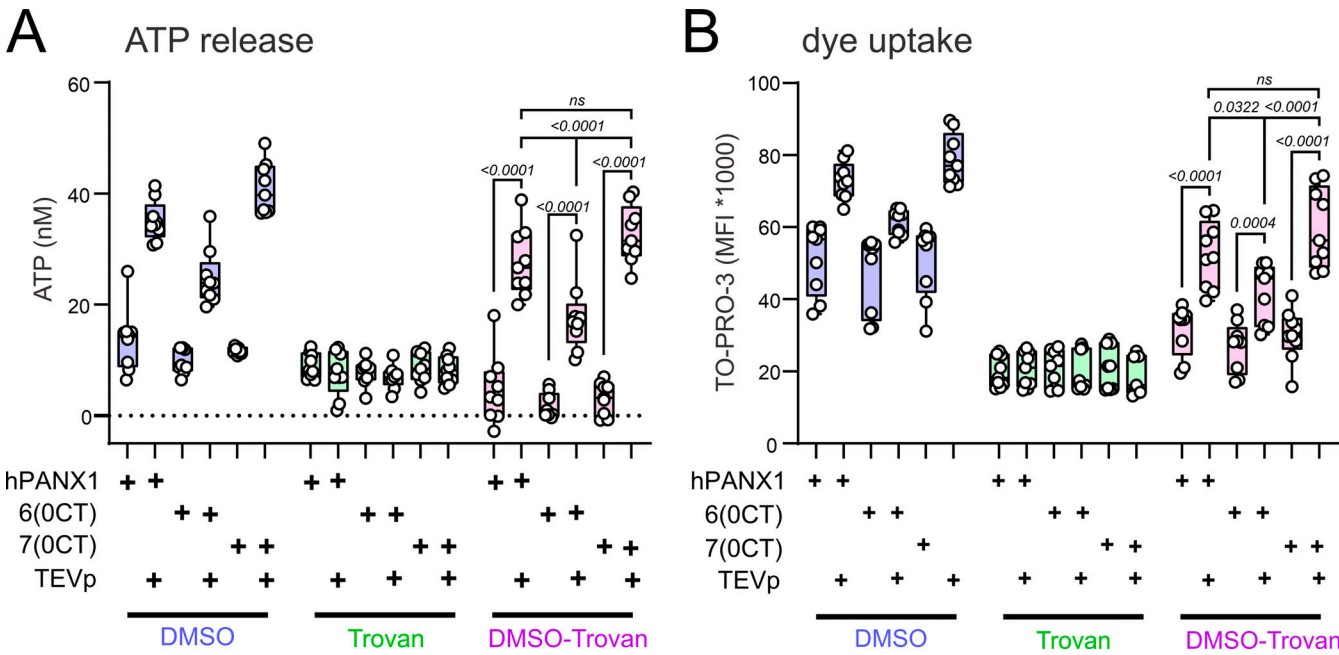

Figure 6. **ATP release and dye uptake is supported more effectively by heptameric versus hexameric hPANX1 concatemers.** hPANX1-deleted HEK293T cells were transfected with hPANX1(TEV) or hexameric and heptameric hPANX1 concatemers along with either TEVp or empty vector and maintained in Trovan (25 μM) following transfection to block hPANX1 channel activity. **(A and B)** The cells were washed and incubated in Trovan or vehicle (DMSO) before assessing ATP release by luciferase assay (A) or TO-PRO-3 dye uptake by flow cytometry (B). The TEVp-activated ATP release and dye uptake that was Trovan-dependent (DMSO-Trovan) was considered to be hPANX1-mediated. Analyses by two-way ANOVA with Šídák's multiple comparisons test.

was smaller and the inward conductance was slightly larger relative to previous results obtained from single subunits or hexameric concatemers in HEK293 cells (~95 and ~12 pS [Chiu et al., 2014, 2017, 2018, 2021; Poon et al., 2014]), yielding a less rectifying unitary i-V relationship. The reason for these differences in conductance under similar recording conditions is not certain but may be associated with the specific PANX1-deleted cell line used here. Notably, we never encountered any of the much larger conductance channels (~500 pS) that have occasionally been reported for PANX1 (Wang et al., 2014; Chiu et al., 2018; Dahl, 2018; Mim et al., 2021).

Also, similar to previous recordings, we found no voltage dependence in $P_O$ of cleavage-activated hPANX1 channels formed from concatenated or single subunit constructs (Chiu et al., 2017, 2018). However, we did note a prominent difference in mean channel open time, with much shorter openings at negative membrane potentials; this was counterbalanced by more frequent openings at those negative potentials to attain the same $P_O$ as at positive potentials. Thus, this analysis revealed a heretofore unrecognized voltage dependence in the channel gating kinetics, albeit one that does not affect the $P_O$ of the channels. Notably, the release of ATP and other metabolites from non-excitable cells will primarily occur at negative membrane potentials, where conductance and open times are the smallest.

**Distinct channel conformations are compatible with active PANX1 gating**

Despite striking advances in the structural biology of PANX1 channels, the specific open or closed conformation of the resolved

channels has remained enigmatic (reviewed in Syrjanen et al. [2021]). Specifically, key features of the permeation pathway appear essentially identical whether the channel is intact or the C-terminus is deleted/truncated at the caspase cleavage site, with a pore constriction that appears compatible with the passage of atomic ions but not large molecules (e.g., ATP, various dyes). Interestingly in this respect, recent work suggests that ionic current can be dissociated from dye permeation in PANX1, CALHM1, and some connexin channels, suggesting the possibility of alternative pathways for these two transport functions in large pore channels (Gaete and Contreras, 2020; Gaete et al., 2020; Nielsen et al., 2020; Gaete et al., 2024). Further complicating the interpretation of the structures, PANX1 channels transition between open and closed states when they are activated by C-tail cleavage (Chiu et al., 2014, 2017; Poon et al., 2014). Our data suggest that this cleavage-activated channel gating, for both ionic currents and dye permeation, requires greater freedom of movement for the N- and/or C-terminal regions of the channel than is allowed by the intersubunit linkers in the concatemers. In this regard, the disposition of these regions is not consistently visible in available PANX1 structures. However, it was recently proposed that, after C-tail cleavage, a C-terminal "channel-activating domain" displaces an N-terminal domain from a common binding pocket, resulting in an N-terminal translocation to a "flipped-up" position in the pore that favors channel opening (Henze et al., 2024). Notably, and by contrast, we found that α1DAR-mediated activation of the concatemers, with intact or truncated C-tails, does not require severing the linkers. From previous work, it is clear that GPCR-activated PANX1 channels

have a similar peak unitary conductance as cleavage-activated, albeit with more "flickery" open-closed kinetics (Chiu et al., 2017). To the extent that a similar N-terminal translocation is also required for channel activation by GPCR-evoked post-translational modifications, as recently proposed (Henze et al., 2024), it may be that the linked N- and C-termini in the concatemers interfere with the ability of the cleaved C-terminal domain to displace the bound N-terminus while not hindering access of the N-terminus to the "flipped-up" position associated with channel activation. It will be interesting to capture PANX1 channels in defined open-closed states, with the disposition of both the N- and/or C-terminal regions visualized, to understand how those regions can differentially influence gating when activated either by C-tail cleavage or GPCR mechanisms.

In conclusion, we present evidence that PANX1 is capable of forming functional channels in either a hexameric or heptameric configuration and show that these two conformations are associated with distinct single-channel properties that may underlie their different propensities for the release of signaling metabolites, like ATP. A definitive confirmation of hexameric hPANX1 channel configuration awaits a high-resolution structural determination, as recently achieved for the alternative stoichiometries of TRPV3 and 5HT3A channels (Lansky et al., 2023; Introini et al., 2024). Likewise, additional structural work may reveal different conformations for cleavage- and receptor-activated channels. Finally, it will be important to identify specific (patho)physiological contexts that involve any of these particular PANX1 channel conformations.

### Data availability

The data are readily available from the corresponding author upon reasonable request.

## Acknowledgments

Crina M. Nimigean served as editor.

The authors thank Drs. Susan Leonhardt and Mark Yeager (University of Miami, Coral Gables, FL, USA) for providing our stock of purified TEV protease, and Drs. Chris Medina and Adishesh Narahari for preliminary experiments of dye uptake in concatemer-expressing cells.

Funding was provided by 113-2628-B-007-002-MY3 from the National Science and Technology Council, Taiwan (to Y.-H. Chiu) and by P01 HL120840 (to B.N. Desai, K.S. Ravichandran, and D.A. Bayliss). Open Access funding provided by the VIVA consortium.

Author contributions: S. Gupta: Conceptualization, Data curation, Formal analysis, Investigation, Methodology, Project administration, Resources, Software, Validation, Visualization, Writing - original draft, Writing - review & editing, Y.-H. Chiu: Conceptualization, Methodology, Resources, Writing - review & editing, M.C. Manjegowda: Formal analysis, B.N. Desai: Funding acquisition, Investigation, Resources, K.S. Ravichandran: Conceptualization, Funding acquisition, Methodology, Project administration, Resources, Supervision, Writing - review & editing, D.A. Bayliss: Conceptualization, Formal analysis, Funding acquisition, Methodology, Project administration, Resources, Supervision, Validation, Visualization, Writing - original draft, Writing - review & editing.

Disclosures: The authors declare no competing interests exist.

Submitted: 26 September 2024

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

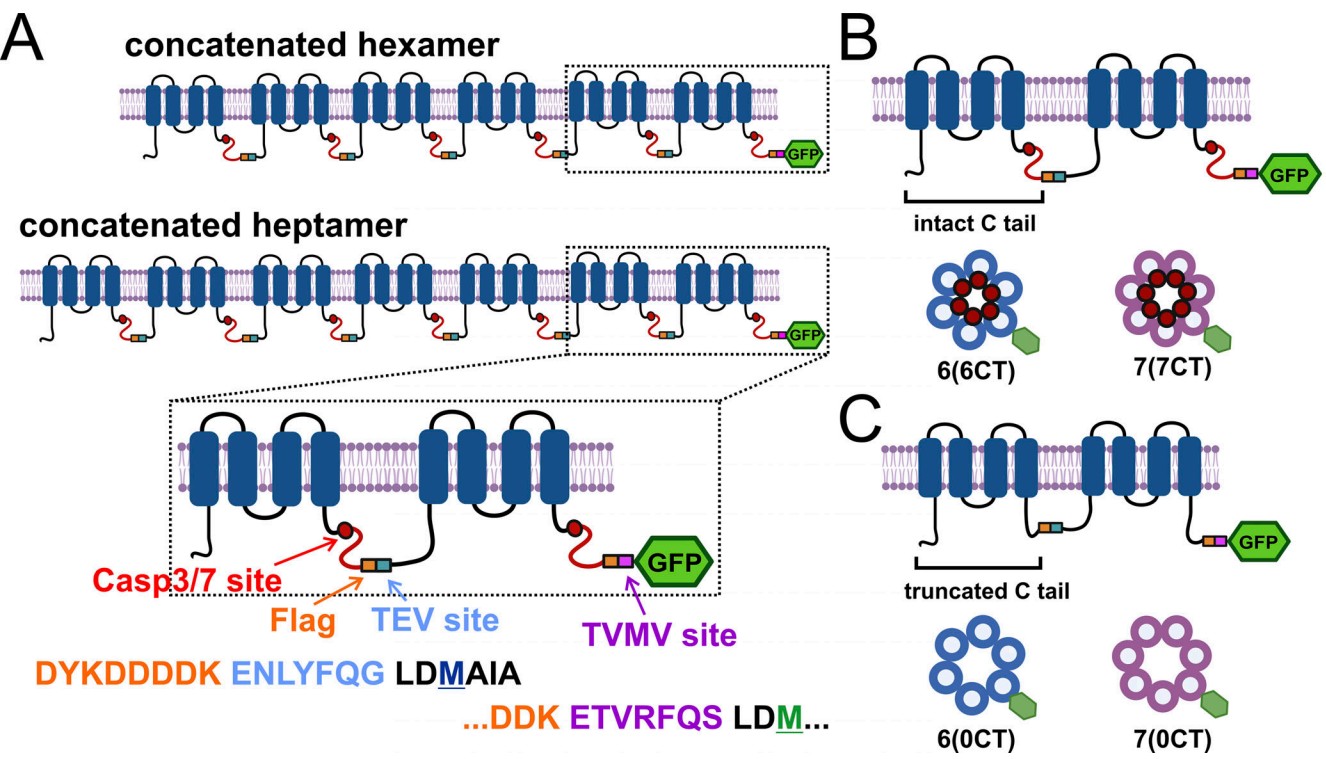

Figure S1. **Schematic of hPANX1 concatemer design. (A)** Illustration of hexameric and heptameric hPANX1 concatemers with intact C-tails. An expanded view of the final two subunits shows the retained C-terminal caspase site and the linker, containing a Flag epitope sequence and a site for TEVp-mediated cleavage; a C-terminal GFP is appended to the last subunit in the concatemers. **(B and C)** Schematized views of final two subunits and expected conformations of 6(6CT) and 7(7CT) hPANX1 concatemers with intact C-tails (B) and of 6(0CT) and 7(0CT) concatemers with C-tails truncated at the C-terminal caspase site (C).

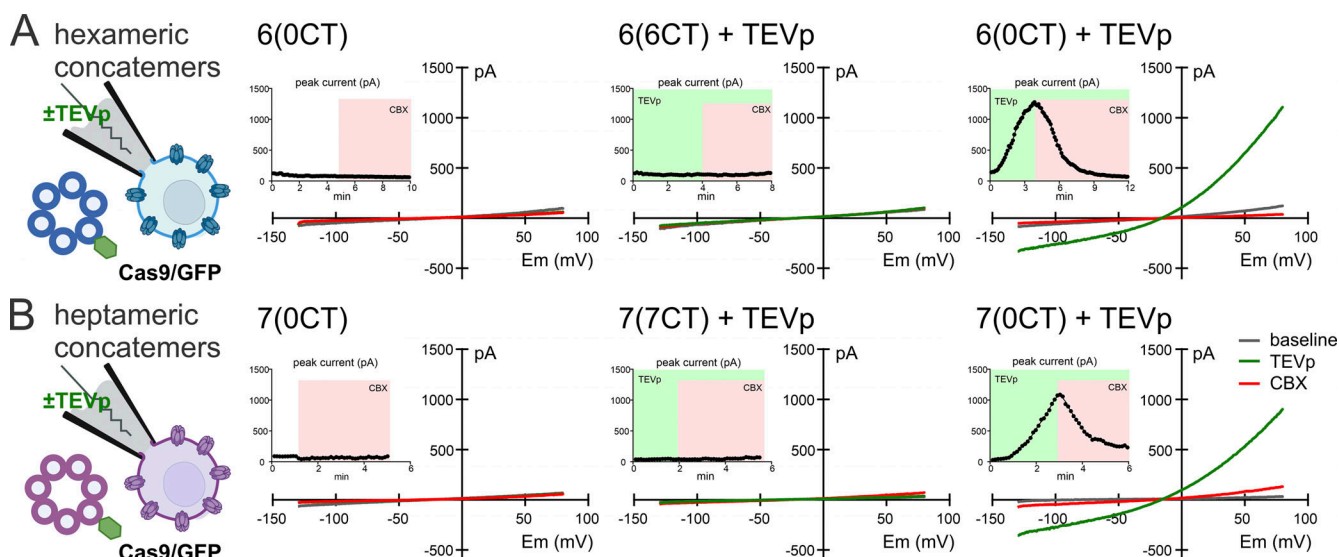

Figure S2. **Hexameric and heptameric concatemers generate comparable whole cell currents in control cells. (A and B)** Schematic of configuration for whole-cell recording of Cas9/GFP-expressing HEK293T cells with TEVp in the pipette to cleave subunit linkers of concatenated hexameric (A) and heptameric (B) PANX1 channels; exemplar ramp *I*-*V* curves and time series of peak current at +80 mV (insets) from the indicated constructs and conditions. Note that CBX-sensitive hPANX1-like currents developed only in cells expressing 6(0CT) and 7(0CT) constructs with truncated C-tails, and only when the pipette contained TEVp to cut the intersubunit linkers.

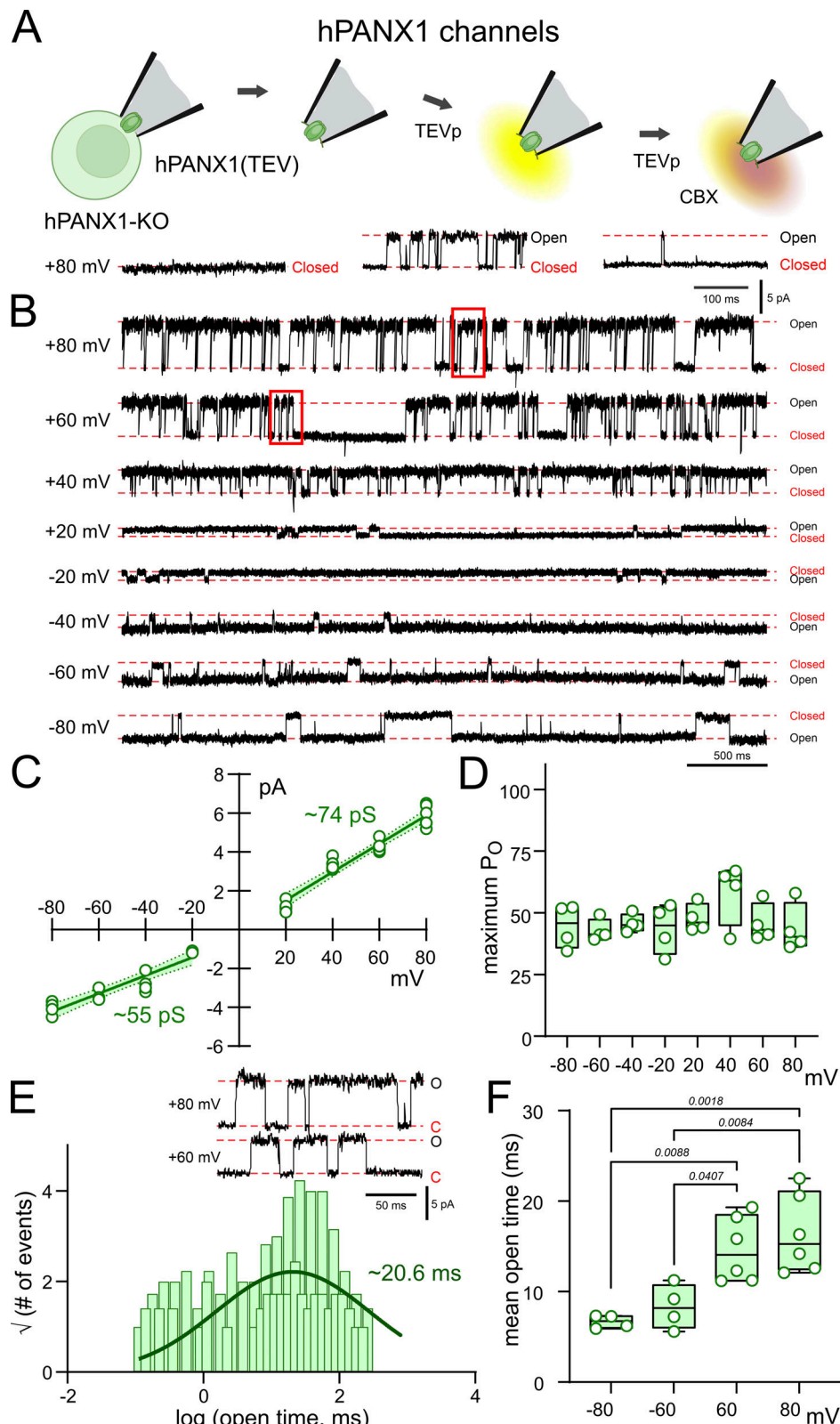

Figure S3. **Unitary hPANX1 channel properties in hPANX1-deleted HEK293T cells. (A)** Inside-out patch recordings of TEVp-activated CBX-sensitive channels from hPANX1-deleted HEK293T cells expressing hPANX1(TEV) channels. **(B)** Recordings of TEVp-activated channels at the indicated membrane potentials. **(C)** Unitary *i*-V relationship for hPANX1 channels, with estimated conductance from the fitted linear regression at positive and negative potentials (shaded regions is the 95% confidence interval). **(D)** Channel $P_O$ at the indicated membrane potentials. **(E)** Open time distribution fitted with a Gaussian function to estimate the channel mean open time. Insets: Expanded channel records at a faster time base (from region indicated in B). **(F)** Channel mean open times at the indicated membrane potentials. Statistical analyses by ANOVA with Tukey's multiple comparisons test.

