## [Peer Review File · The Journal of General Physiology]

Distinct properties and activation of hexameric and heptameric Pannexin 1 channel concatemers

Smriti Gupta, Yu-Hsin Chiu, Mohan Manjegowda, Bimal Desai, Kodi Ravichandran, and Douglas Bayliss

Corresponding Author(s): Douglas Bayliss, University of Virginia Health System

Review Timeline:

Submission Date:	September 26, 2024
Editorial Decision:	October 29, 2024
Revision Received:	November 12, 2024
Editorial Decision:	November 20, 2024
Revision Received:	November 25, 2024

Editor: Crina Nimigean

Transaction Report:

DOI: <https://doi.org/10.1085/jgp.202413676>

October 29, 2024

Dr. Douglas A Bayliss
University of Virginia Health System
Dept. of Pharmacology
PO Box 800735
Charlottesville, VA 22908

Re: 202413676

Dear Dr. Bayliss,

Thank you for submitting your manuscript, entitled "Activation and Unitary Properties of Hexameric and Heptameric Pannexin 1 Channel Concatemers" to JGP. Your manuscript has now been seen by 3 reviewers, whose comments are appended below. You will see that the reviewers were overall very enthusiastic about the study and its potential impact and raised minor concerns that should nevertheless be addressed prior to further consideration of the manuscript at JGP. Of note, you will see that both reviewers 2 and 3 would like to see structural evidence of the concatemers' stoichiometry. The editors agree that, although structural evidence is not required for the publication, it would significantly strengthen the message of the paper if included.

We hope that you will be able to submit a revised manuscript that addresses these points, which we believe will pose no problems, and which may be re-reviewed. In addition, please do not hesitate to contact me (via the editorial office) if you feel that a discussion of the reviewers' and editors' comments would be helpful.

Please submit your revised manuscript via the link below, along with a point-by-point letter that details your response to the reviewers' and editors' comments, as well as a copy of the text with alterations highlighted (boldfaced or underlined). If the article is eventually accepted, it would include a 'revised date' as well as submitted and accepted dates. If we do not receive the revised manuscript within one year, we will regard the article as having been withdrawn. We would be willing to receive a revision of the manuscript at a later time, but the manuscript will then be treated as a new submission, with a new manuscript number.

Please pay particular attention to recent changes to our instructions to authors in the following sections: Data presentation, Blinding and randomization and Statistical analysis, under Materials and Methods, as shown here: <https://rupress.org/jgp/pages/submission-guidelines#prepare>. Re-review will be contingent on inclusion of the required information (including for data added during revision) and demonstration of the experimental reproducibility of the results. Also, To improve the reproducibility of published content, we have partnered with SciScore. Authors are prompted in eJP to copy and paste the Materials and Methods section of their manuscript for a SciScore assessment when submitting their revised manuscript. Authors are encouraged (not required) to further revise their Materials and Methods if the SciScore is below 4. More information can be found here: <https://rupress.org/jgp/pages/submission-guidelines#sciscore>.

Please note, JGP now requires authors to submit Source Data used to generate figures containing gels and Western blots with all revised manuscripts (when applicable). This Source Data consists of fully uncropped and unprocessed images for each gel/blot displayed in the main and supplemental figures. If your paper includes cropped gel and/or blot images, please be sure to provide one Source Data file for each figure that contains gels and/or blots along with your revised manuscript files. File names for Source Data figures should be alphanumeric without any spaces or special characters (i.e., SourceDataF#, where F# refers to the associated main figure number or SourceDataFS# for those associated with Supplementary figures). The lanes of the gels/blots should be labeled as they are in the associated figure, the place where cropping was applied should be marked (with a box), and molecular weight/size standards should be labeled wherever possible. Source Data files will be made available to reviewers during evaluation of revised manuscripts and, if your paper is eventually published in JGP, the files will be directly linked to specific figures in the published article.

Source Data Figures should be provided as individual PDF files (one file per figure). Authors should endeavor to retain a minimum resolution of 300 dpi or pixels per inch. Please review our instructions for export from Photoshop, Illustrator, and PowerPoint here: <https://rupress.org/jgp/pages/submission-guidelines#revised>

Whilst you are revising your manuscript, we ask that you consider whether you have any artwork that might be suitable for the cover of JGP. Microscopy images are particularly good for cover artwork, but other types of image can be very effective, so we encourage you to be creative. Please don't restrict yourself to images from the paper; an image that is relevant to the work described would be just as suitable. Images should be a minimum resolution of 300 dpi. To see recent examples, visit the following page and click on 'Show covers? Yes': <https://jgp.rupress.org/content/by/year>

Thank you for submitting your interesting research to JGP.

Please submit your revised manuscript, and any associated files, via this link:
Link Not Available

Sincerely,

Crina Nimigean, Ph.D.
On behalf of Journal of General Physiology

Journal of General Physiology's mission is to publish mechanistic and quantitative molecular and cellular physiology of the highest quality; to provide a best-in-class author experience; and to nurture future generations of independent researchers.

Reviewer #1 (Comments to the Authors):

The goal of this paper is to resolve confusion regarding the functional properties of Panx1 channels and their relation to the number of subunits that form the channels. Previous work by this group contributed to that controversy. This paper does a masterful job of addressing the issue fully, comprehensively, and conclusively. Along the way, additional and intriguing functional information about Panx1 channel activation and single channel properties is obtained.

Previous work by this group (Chiu et al., 2017) provided what seemed like solid data on the properties of hexameric Panx1 channels at a time when it was presumed, with some basis, that the native channels were hexameric, and also that homomultimeric channels always assembled with a single stoichiometry (and that it was always an even number).

It has since become clear that neither presumption is true, and specifically, that native Panx1 channels assemble as heptamers. The confusing finding from the 2017 work was that the "hexameric" channels (which were derived from sets 2- and 3-mer concatemers) had unitary conductance matching that of Panx1 channels formed by non-concatemeric Panx1 (i.e., the native structures), now believed to be heptameric.

The present work delves deeply and thoroughly into this conundrum. The two key innovations in this work are (a) expression of Panx1 in cells in which any potential, yet undetected, Panx1 expression was deleted by CRISPR/Cas9 (and also in cells in which it was not, as in the original study), and (b) rather than express concatenated 2- and 3-mers of Panx1 as in the previous work and rely on secondary data to support hexameric assembly, here the Panx1 is expressed as concatenated 6-, 7-mers or single subunits. The concatemers contained TEVp cleavage sites between the monomers.

The three constructs - singly expressed monomers and each concatemer - were expressed in each of the two types of cells, before and after TEV cleavage, before and after C-terminal caspase cleavage (which activates the channels), and before and after exposure to phenylephrine (PE) in cells also expressing the α_1 DAR. Every base was covered. Data were obtained regarding macroscopic currents and their I-V relations, single channel currents, open/closed dwell-times and Pos, and molecular permeability.

The motivation for each permutation of conditions in pursuit of an explanation is described in detail. The results are clear: Panx1 can form caspase-cleavage-activated hexamers or heptamers, after TEV cleavage of concatemers. However, the unitary conductances and the single-channel I-V relations of the heptamers correspond to those of Panx1 channels formed Panx1 expressed as monomers, and those of the hexamers do not. So hexamers can indeed be functional, but Panx1 is expressed natively as heptamers.

I believe the authors correctly infer that the unitary conductances seen in the 2017 paper attributed to hexameric channels were due to (undetected) endogenous expression of Panx1 monomers that rapidly equilibrated with the apparently hexameric channels, following TEV cleavage, to form heptamers. It is mentioned that rapid transitions between different multimeric states has been seen for TRPV3 channels. The results in this paper are consistent with this explanation, but, surprisingly, the authors never explicitly state this as the resolution of the motivating issue of this work. This should be made explicit if it is indeed the explanation of the previous work.

This study also illustrates an important cautionary note about the use of cleavable concatemeric constructs for other multimeric channels - the possibility of rapid changes in multimeric form that have functional consequences, a consequence of the now accepted idea that these kinds of channels can have different functional multimeric forms.

The studies also make several new points about Panx1 channels. At a macroscopic level, the most surprising and intriguing was the finding that PE was able to activate the concatemeric Panx1 channels *before* the TEV protease unlinked the component monomers. This was unexpected since channel activity mediated by linked Panx1 monomers has not been seen previously. This suggests that GPCR-mediated Panx1 activation is by a different means/mechanism than that induced by C-terminal caspase cleavage. At the single channel level, the unitary conductances of Panx1 were found to be rectifying, with stronger rectification

was stronger in the hexameric channels.

Overall, this puts the stoichiometry/unitary conductance controversy to rest and provides an expert characterization of the channels, adding information about Panx1 channel activation by GPCRs that would not have been obtained without the investigation of concatemeric forms of the channel. The thoroughness with which this issue was investigated is impressive and well-described.

There are some language/expression issues that caused me some initial confusion:

For example:

Line 156: "However, the 6(OCT) channels recorded here had a smaller outward conductance (~55 pS) than observed previously for either wild-type hPANX1 or 6(OCT) constructs in HEK293T cells (~95 pS) Chiu et al., 2017), *likely from some cell-specific effect of the CRISPR/Cas9 targeting/drug selection process*." [my emphasis]

I believe they are saying that this difference is due to the deletion of endogenous Panx1 - if so, that should be stated rather than saying is it "some cell-specific effect of the CRISPR/Cad9". If this is not what is intended, I do not know what the phrase is referencing.

Another:

Line 279: "...the consistent and distinct single channel properties obtained from the hexameric and heptameric concatemers suggests that our recordings did not arise from a single population of channels of either stoichiometry, and are most consistent with faithful production of channels in both hexameric and heptameric conformations."

This is confusing - it implies that channels of both conductances were seen when either the 6-mer or the 7-mer were expressed, or maybe that there was an experiment in which 6-mers and 7-mers were expressed in the same cells. I don't think either was the case, and am unsure what this statement means.

Abstract:

"mean channel open times similar to that from an unlinked single Panx1 subunit"

I read this to indicate that a single Panx1 subunit made a channel....

"In addition, the [TEV-cleaved] heptameric concatemers supported...."

Line 35: "assembled from [expression of] single Panx1 subunits"

Line 330: these studies are not about CALHM1 channels, but are about pannexin channels.

Reviewer #2 (Comments to the Authors):

The paper titled "Activation and Unitary Properties of Hexameric and Heptameric Pannexin 1 Channel Concatemers" by Smriti Gupta et al analyzes concatenated PANX1 in 6-mer and 7-mer forms, suggesting that PANX1 could form a hexamer based on electrophysiological data and ATP/dye release assays. A similar approach was employed by the Bayliss lab in their 2017 Nature Communications paper (PMID: 28134257), which also concluded that PANX1 is a hexamer. However, this model has been challenged by later high-resolution cryo-EM studies, all of which demonstrated a heptameric stoichiometry for PANX1 channel, including a preprint from the same group of researchers in this study (<https://doi.org/10.1101/2020.12.31.425019>). Currently, it is widely accepted that PANX1 forms a heptamer, not a hexamer. While I have no doubts about the technical execution of this study, I find the research design and results presented in this manuscript less convincing. It's really difficult to affirm channel stoichiometry with confidence from electrophysiology analysis alone. As the authors noted in the discussion, it is likely that after TEV cleavage, the channel subunits re-folded/shuffled into their preferred stoichiometry, which could account for the observed currents. The difference in terms of single-channel properties could be due to the specific constructs (6-mers/7-mers) in protein folding, expression level, trafficking/processing, and kinetics of assembly/shuffling after TEV treatment. To definitively prove that PANX1 can form a hexamer, the authors should employ electron microscopy to analyze the 6-mer constructs (with and without TEV treatment), and see if the 2D class averages could reveal a hexameric structure. In all instances where unusual stoichiometries, such as with 5-HT3, TRPV3, and CALHM proteins, have been observed, electron microscopy was used to confirm the arrangement. Therefore, such evidence is essential to support the main conclusion of this paper.

Reviewer #3 (Comments to the Authors):

The work by Gupta et al describes assesses the single channel properties of hexameric and heptameric pannexin channel (PANx1). PANX1 is a family of large pore membrane ion channels that facilitate the conductance of ions and signaling molecules such as ATP and fluorescent dyes. Although initial evidence suggested PANX1 formed as a hexamer, a series of cryo-EM

studies have shown a heptameric assembly. To investigate whether hexameric and heptameric forms support PANX1 channel activity, the authors studied PANX1 constructs comprising 6 or 7 subunits with intact or truncated C-termini, mimicking caspase-cleavage activation. Whole-cell recordings from PANX1-deleted cells revealed that both hexameric and heptameric concatemers with truncated C-tails generated PANX1-like currents after the inter-subunit linkers were cleaved by TEV protease at the engineered site. Interestingly, stimulation of α 1D adrenoceptors activated the constructs regardless of whether the linkers were cleaved. In inside-out patch recordings, cleaved heptameric channels showed higher conductance and longer open times compared to hexameric channels. Moreover, heptameric concatemers facilitated greater ATP release and TO-PRO-3 uptake than their hexameric counterparts. The authors concluded that PANX1 can function in both hexameric and heptameric conformations, with the heptameric form facilitating large molecule permeation more effectively. The data also suggest that PANX1 activation involves distinct structural mechanisms for C-tail cleavage and receptor-mediated activation. The experiments are well-designed, and the authors appropriately address the potential caveats and limitations of their work in the discussion. This work deserves to be published with minor revisions. I have a few specific comments:

- The message of the manuscript could be significantly strengthened if the authors can demonstrate that the hexameric and heptameric concatemers indeed form hexamers and heptamers, respectively. If the research environment is set up to perform negative stain EM, as in Chiu et al. (2017), this would be exciting to see. While this is not required for acceptance of the paper, it could potentially be a valuable addition to the authors' future work.
- The design of the concatemer constructs is sophisticated, and a schematic figure panel would greatly benefit the manuscript.
- The authors should consider speculating on how Panx1 might be activated by GPCRs. The finding that the concatemer without TEV cleavage remains active when co-expressed with alpha-adrenergic receptors is fascinating. While I'm sure the authors are following up on this, it would be helpful to include their working hypotheses in the Discussion section.
- Line 106: The statement "G protein-coupled receptor activation of PANX1 proceeds in hexameric and heptameric constructs even with intact inter-subunit linkers" is unclear. The authors could clarify this subheading.
- Line 22: There are six CALHM members (CALHM1-6), but if the authors are referring to functional channels, only CALHM1 homomers or CALHM1/3 heteromers have been shown to exhibit channel activity (with consensus).
- Line 24: The authors could mention that CALHM1 (Syrjanen et al., NSMB 2020) is an outlier concerning membrane topology.

We thank the reviewers for the positive assessment of our work and their constructive suggestions for improvement. Below, please find a point-by-point response to the comments.

Reviewer #1 (Comments to the Authors):

The goal of this paper is to resolve confusion regarding the functional properties of Panx1 channels and their relation to the number of subunits that form the channels. Previous work by this group contributed to that controversy. This paper does a masterful job of addressing the issue fully, comprehensively, and conclusively. Along the way, additional and intriguing functional information about Panx1 channel activation and single channel properties is obtained.

Previous work by this group (Chiu et al., 2017) provided what seemed like solid data on the properties of hexameric Panx1 channels at a time when it was presumed, with some basis, that the native channels were hexameric, and also that homomultimeric channels always assembled with a single stoichiometry (and that it was always an even number).

It has since become clear that neither presumption is true, and specifically, that native Panx1 channels assemble as heptamers. The confusing finding from the 2017 work was that the "hexameric" channels (which were derived from sets 2- and 3-mer concatemers) had unitary conductance matching that of Panx1 channels formed by non-concatemeric Panx1 (i.e., the native structures), now believed to be heptameric.

The present work delves deeply and thoroughly into this conundrum. The two key innovations in this work are (a) expression of Panx1 in cells in which any potential, yet undetected, Panx1 expression was deleted by CRISPR/Cas9 (and also in cells in which it was not, as in the original study), and (b) rather than express concatenated 2- and 3-mers of Panx1 as in the previous work and rely on secondary data to support hexameric assembly, here the Panx1 is expressed as concatenated 6-, 7-mers or single subunits. The concatemers contained TEVp cleavage sites between the monomers.

The three constructs - singly expressed monomers and each concatemer - were expressed in each of the two types of cells, before and after TEV cleavage, before and after C-terminal caspase cleavage (which activates the channels), and before and after exposure to phenylephrine (PE) in cells also expressing the alpha1DAR. Every base was covered. Data were obtained regarding macroscopic currents and their I-V relations, single channel currents, open/closed dwell-times and Pos, and molecular permeability.

The motivation for each permutation of conditions in pursuit of an explanation is described in detail. The results are clear: Panx1 can form caspase-cleavage-activated hexamers or heptamers, after TEV cleavage of concatemers. However, the unitary conductances and the single-channel I-V relations of the heptamers correspond to those of Panx1 channels formed Panx1 expressed as monomers, and those of the hexamers do not. So hexamers can indeed be functional, but Panx1 is expressed natively as heptamers.

I believe the authors correctly infer that the unitary conductances seen in the 2017 paper attributed to hexameric channels were due to (undetected) endogenous expression of Panx1 monomers that rapidly equilibrated with the apparently hexameric channels, following TEV cleavage, to form heptamers. It is mentioned that rapid transitions between different multimeric states has been seen for TRPV3 channels. The results in this paper are consistent with this explanation, but, surprisingly, the authors never explicitly state this as the resolution of the motivating issue of this work. This should be made explicit if it is indeed the explanation of the previous work.

This study also illustrates an important cautionary note about the use of cleavable concatemeric constructs for other multimeric channels - the possibility of rapid changes in multimeric form that have functional consequences, a consequence of the now accepted idea that these kinds of channels can have different functional multimeric forms.

The studies also make several new points about Panx1 channels. At a macroscopic level, the most surprising and intriguing was the finding that PE was able to activate the concatemeric Panx1 channels *before* the TEV protease unlinked the component monomers. This was unexpected since channel activity mediated by linked Panx1 monomers has not been seen previously. This suggests that GPCR-mediated Panx1 activation is by a different means/mechanism than that induced by C-terminal caspase cleavage. At the single channel level, the unitary conductances of Panx1 were found to be rectifying, with stronger rectification was stronger in the hexameric channels.

Overall, this puts the stoichiometry/unitary conductance controversy to rest and provides an expert characterization of the channels, adding information about Panx1 channel activation by GPCRs that would not have been obtained without the investigation of concatemeric forms of the channel. The thoroughness with which this issue was investigated is impressive and well-described.

Thank you for this detailed, thoughtful, and constructive review.

There are some language/expression issues that caused me some initial confusion:

For example:

Line 156: "However, the 6(OCT) channels recorded here had a smaller outward conductance (~55 pS) than observed previously for either wild-type hPANX1 or 6(OCT) constructs in HEK293T cells (~95 pS) Chiu et al., 2017), *likely from some cell-specific effect of the CRISPR/Cas9 targeting/drug selection process*." [my emphasis]

I believe they are saying that this difference is due to the deletion of endogenous Panx1 - if so, that should be stated rather than saying is it "some cell-specific effect of the CRISPR/Cad9". If this is not what is intended, I do not know what the phrase is referencing.

We have replaced the original vague clause with the following sentence (l. 158-160):

"We suspect that properties attributed to the hexameric channels following TEV cleavage in that earlier work may have arisen instead from heptameric channels formed after rapid incorporation of endogenous hPANX1 monomers into the cleaved hexameric constructs."

We trust that this explanation of the previous work is now more explicit, as also requested in the general comments above.

Another:

Line 279: "...the consistent and distinct single channel properties obtained from the hexameric and heptameric concatemers suggests that our recordings did not arise from a single population of channels of either stoichiometry, and are most consistent with faithful production of channels in both hexameric and heptameric conformations."

This is confusing - it implies that channels of both conductances were seen when either the 6-mer or the 7-mer were expressed, or maybe that there was an experiment in which 6-mers and 7-mers were expressed in the same cells. I don't think either was the case, and am unsure what this statement means.

Thank you for pointing out this confusion; indeed, neither was the case.

We have now re-worded that sentence to say (l. 281-284): "... *the consistent and distinct single channel properties obtained, respectively, from expressing either the hexameric or heptameric concatemers are most consistent with recordings derived from faithful production of channels in the expected hexameric and heptameric conformations.*"

Abstract:

"mean channel open times similar to that from an unlinked single Panx1 subunit"

I read this to indicate that a single Panx1 subunit made a channel....

Thank you for pointing out this ambiguity. This now reads: "*similar to channels assembled from expression of unlinked single PANX1 subunits.*"

"In addition, the [TEV-cleaved] heptameric concatemers supported...."

We added "linker-cleaved" to the indicated sentence. (We did not specify the protease used to cleave the linker because "TEV" is not yet defined at this point in the manuscript.)

Line 35: "assembled from [expression of] single Panx1 subunits"

l. 37: Added "*expression of ...*", per suggestion.

Line 330: these studies are not about CALHM1 channels, but are about pannexin channels.

l. 331: We added reference to PANX1 in this sentence. We retained mention of CALHM1 because, by our reading, it was included in the work in Gaete et al., 2020 (J Gen Physiol, 152: e202012607) in which a dissociation of YO-PRO uptake and ionic current at negative membrane potentials was observed in *Xenopus* oocytes expressing CALHM1.

Reviewer #2 (Comments to the Authors):

The paper titled "Activation and Unitary Properties of Hexameric and Heptameric Pannexin 1 Channel Concatemers" by Smriti Gupta et al analyzes concatenated PANX1 in 6-mer and 7-mer forms, suggesting that PANX1 could form a hexamer based on electrophysiological data and ATP/dye release assays. A similar approach was employed by the Bayliss lab in their 2017 Nature Communications paper (PMID: 28134257), which also concluded that PANX1 is a hexamer. However, this model has been challenged by later high-resolution cryo-EM studies, all of which demonstrated a heptameric stoichiometry for PANX1 channel, including a preprint from the same group of researchers in this study (<https://doi.org/10.1101/2020.12.31.425019>). Currently, it is widely accepted that PANX1 forms a heptamer, not a hexamer. While I have no doubts about the technical execution of this study, I find the research design and results presented in this manuscript less convincing. It's really difficult to affirm channel stoichiometry with confidence from electrophysiology analysis alone. As the authors noted in the discussion, it is likely that after TEV cleavage, the channel subunits re-folded/shuffled into their preferred stoichiometry, which could account for the observed currents. The difference in terms of single-channel properties could be due to the specific constructs (6-mers/7-mers) in protein folding, expression level, trafficking/processing, and kinetics of assembly/shuffling after TEV treatment. To definitively prove that PANX1 can form a hexamer, the authors should employ electron microscopy to analyze the 6-mer constructs (with and without TEV treatment), and see if the 2D

class averages could reveal a hexameric structure. In all instances where unusual stoichiometries, such as with 5-HT₃, TRPV3, and CALHM proteins, have been observed, electron microscopy was used to confirm the arrangement. Therefore, such evidence is essential to support the main conclusion of this paper.

We thank the referee for positive comments on the technical execution of the study. We also appreciate the value of providing structural validation for the proposed stoichiometry. Unfortunately, we are not currently positioned to perform those studies.

As pointed out by the referee, we indeed acknowledge in the Discussion a caveat that some reshuffling of subunits is theoretically possible from channels already on the cell membrane, even within seconds to minutes of TEVp application. However, the single TEVp-activated channels derived from expressing either the hexameric and heptameric concatemers, respectively, showed consistent and distinctive properties. We think this provides strong support that each of those concatemers generated a unique population of channels in the expected stoichiometric conformation. Note also that these surface localized channels will have already undergone protein folding and membrane trafficking, and that channel unitary properties are independent of expression levels.

In sum, we consider it extremely unlikely that subunits freed from the hexameric and heptameric concatemers would generate some preferred (heptameric) conformation that somehow present with distinctively different unitary properties dictated by the original concatemeric construct from which the subunits were liberated.

Reviewer #3 (Comments to the Authors):

The work by Gupta et al describes and assesses the single channel properties of hexameric and heptameric pannexin channel (PANx1). PANX1 is a family of large pore membrane ion channels that facilitate the conductance of ions and signaling molecules such as ATP and fluorescent dyes. Although initial evidence suggested PANX1 formed as a hexamer, a series of cryo-EM studies have shown a heptameric assembly. To investigate whether hexameric and heptameric forms support PANX1 channel activity, the authors studied PANX1 constructs comprising 6 or 7 subunits with intact or truncated C-termini, mimicking caspase-cleavage activation. Whole-cell recordings from PANX1-deleted cells revealed that both hexameric and heptameric concatemers with truncated C-tails generated PANX1-like currents after the inter-subunit linkers were cleaved by TEV protease at the engineered site. Interestingly, stimulation of α 1D adrenoceptors activated the constructs regardless of whether the linkers were cleaved. In inside-out patch recordings, cleaved heptameric channels showed higher conductance and longer open times compared to hexameric channels. Moreover, heptameric concatemers facilitated greater ATP release and TO-PRO-3 uptake than their hexameric counterparts. The authors concluded that PANX1 can function in both hexameric and heptameric conformations, with the heptameric form facilitating large molecule permeation more effectively. The data also suggest that PANX1 activation involves distinct structural mechanisms for C-tail cleavage and receptor-mediated activation.

The experiments are well-designed, and the authors appropriately address the potential caveats and limitations of their work in the discussion. This work deserves to be published with minor revisions. I have a few specific comments:

Thank you for the positive assessment of the experiments and this support for publication of the results.

- The message of the manuscript could be significantly strengthened if the authors can demonstrate that the hexameric and heptameric concatemers indeed form hexamers and heptamers, respectively. If the research environment is set up to perform negative stain EM, as in Chiu et al. (2017), this would be exciting to see. While this is not required for acceptance of the paper, it could potentially be a valuable addition to the authors' future work.

We appreciate this suggestion. Unfortunately, we are not currently in a position to perform those studies. As outlined above (see Reviewer 2), we believe that the distinctive unitary properties of the channels obtained from expressing the two different concatemers is strong evidence that the channels indeed form in the expected stoichiometry.

- The design of the concatemer constructs is sophisticated, and a schematic figure panel would greatly benefit the manuscript.

Thank you for this suggestion. We have included a new supplemental figure with a schematic to illustrate the concatemer design (Supp. Fig. 1), and we provided some additional information in the relevant Methods section.

- The authors should consider speculating on how Panx1 might be activated by GPCRs. The finding that the concatemer without TEV cleavage remains active when co-expressed with alpha-adrenergic receptors is fascinating. While I'm sure the authors are following up on this, it would be helpful to include their working hypotheses in the Discussion section.

We agree that these observations are especially fascinating and we are indeed following up. Specifically, concatemeric channels retaining linked N- and C-termini are incompatible with cleavage-based activation but GPCR-mediated activation can occur even with concatemers that remain linked – and this implies some structural difference in these two forms of activation.

In deference to this request for a speculative explanation, we offer a hypothesis that aligns with a recent mechanistic model proposed for N- and C-terminal interactions underlying channel activation by both C-tail cleavage and GPCR-evoked posttranslational modifications (see Henze et al., 2024). Specifically, we now state (l. 347): *“To the extent that a similar N-terminal translocation is required also for channel activation by GPCR-evoked posttranslational modifications, as recently proposed (Henze et al., 2024), it may be that the linked N- and C-termini in the concatemers interfere with the ability of the cleaved C-terminal domain to displace the bound N-terminus while not hindering access of the N-terminus to the “flipped-up” position that was associated with channel activation.”*

- Line 106: The statement "G protein-coupled receptor activation of PANX1 proceeds in hexameric and heptameric constructs even with intact inter-subunit linkers" is unclear. The authors could clarify this subheading.

We have changed this subheading to (l. 107): *“TEVp-mediated linker cleavage is not required for GPCR activation of PANX1 concatemers.”*

- Line 22: There are six CALHM members (CALHM1-6), but if the authors are referring to functional channels, only CALHM1 homomers or CALHM1/3 heteromers have been shown to exhibit channel activity (with consensus).

l. 20-28: We have now re-worded this section to remove reference to the number of subunits within the different subgroups.

- Line 24: The authors could mention that CALHM1 (Syrjanen et al., NSMB 2020) is an outlier concerning membrane topology.

I. 26: The unique membrane topology of the CALHM1 protomers is now mentioned.

November 21, 2024

Dr. Douglas A Bayliss
University of Virginia Health System
Dept. of Pharmacology
PO Box 800735
Charlottesville, VA 22908

Re: 202413676R1

Dear Dr. Bayliss,

I am pleased to let you know that your manuscript, entitled "Activation and Unitary Properties of Hexameric and Heptameric Pannexin 1 Channel Concatemers" is scientifically acceptable for publication in Journal of General Physiology. Formal acceptance will follow when it is modified in accordance with the referees' remarks and our editorial policies.

Please note items that need attention are listed at the bottom of this email (under 'manuscript formatting checklist') and on the attached marked-up pdf file. Please also be sure to include a letter addressing the reviewers' comments point-by-point (if applicable) and a copy of the text with alterations highlighted (boldfaced or underlined). Your manuscript should be a double-spaced MS Word file and include editable tables, if appropriate.

JGP requires a data availability statement for all research article submissions. These statements will be published in the article directly above the Acknowledgments. The statement should address all data underlying the research presented in the manuscript. Please visit the JGP instructions for authors for guidelines and examples of statements at <https://rupress.org/jgp/pages/editorial-policies#data-availability-statement>.

Please submit your final files via this link:
Link Not Available

Thank you for choosing to publish your research in JGP and please feel free to contact me with any questions.

Sincerely,

Crina Nimigean, Ph.D.
On behalf of Journal of General Physiology

Journal of General Physiology's mission is to publish mechanistic and quantitative molecular and cellular physiology of the highest quality; to provide a best in class author experience; and to nurture future generations of independent researchers.

Manuscript formatting checklist:

- MS Word document of text needed (including editable tables)
 - MS Word document of supplemental text needed, if applicable (including figure legends and editable tables)
 - Brief Statement describing supplementary information needed, if applicable (in subsection at end of Materials & Methods)
 - Please include a data availability statement preceding the Acknowledgments section. Please see <https://rupress.org/jgp/pages/editorial-policies#data-availability-statement>
 - Figures created at sufficient resolution and in acceptable format (including supplemental if applicable). If working in Illustrator, we prefer .ai or .eps file format. If working in Photoshop please use 600dpi/1000dpi .tiff or .psd file format. Minimum resolution at estimated print size: Minimum resolution for all figures is 600 dpi. For figures that contain both photographs and line art or text, 600 dpi is highly recommended. Figures containing only black and white elements (line art, no color, and no gray) should be 1,000 dpi. Maximum figure size is 7 in wide x 9 in high (17.5 x 22.8 cm) at the correct resolution. <https://jgp.rupress.org/fig-vid-guidelines>
 - Supplemental figures, if any, conforming to same guidelines as manuscript figures (noted above)
 - If images resemble one from a prior publications, the author must seek permissions (to reproduce or adapt) from the original publisher. [You can resubmit your paper while waiting to hear back from the original publisher but please keep us updated]
 - All authors must complete a disclosure form prior to acceptance. A link to complete the form has been sent to all coauthors. Please provide the editorial office with updated email addresses if necessary
-

From the marked-up pdf file:

Title changed to “*Distinct properties and activation of hexameric and heptameric Pannexin 1 channel concatemers*”

Summary Statement changed to: “*Gupta et al show that Pannexin 1 (PANX1) channels can function as both hexamers and heptamers, but that heptameric PANX1 shows higher conductance, longer mean open times, and supports greater ATP release and dye uptake. Structurally distinct mechanisms underlie C-tail cleavage and receptor-mediated PANX1 activation.*”

Abstract changed to remove data values.

From the manuscript formatting checklist:

- MS Word document of text needed (including editable tables)

Submitted.

- MS Word document of supplemental text needed, if applicable (including figure legends and editable tables)

Supplemental Figure legends in main manuscript file (p. 25-26, l. 561-586); also included now as a separate DOC file.

- Brief Statement describing supplementary information needed, if applicable (in subsection at end of Materials & Methods)

Statement provided (p. 21, l. 488-492).

- Please include a data availability statement preceding the Acknowledgments section. Please see <https://rupress.org/jgp/pages/editorial-policies#data-availability-statement>

Statement provided (p. 30, l. 700-701).

- Figures created at sufficient resolution and in acceptable format (including supplemental if applicable). If working in Illustrator, we prefer .ai or .eps file format. If working in Photoshop please use 600dpi/1000dpi .tiff or .psd file format. Minimum resolution at estimated print size: Minimum resolution for all figures is 600 dpi. For figures that contain both photographs and line art or text, 600 dpi is highly recommended. Figures containing only black and white elements (line art, no color, and no gray) should be 1,000 dpi. Maximum figure size is 7 in wide x 9 in high (17.5 x 22.8 cm) at the correct resolution. <https://jgp.rupress.org/fig-vid-guidelines>

Figures provided as TIF files at 600 dpi.

- Supplemental figures, if any, conforming to same guidelines as manuscript figures (noted above)

Supplemental Figures provided as TIF files at 600 dpi.

- If images resemble one from a prior publications, the author must seek permissions (to reproduce or adapt) from the original publisher. [You can resubmit your paper while waiting to hear back from the original publisher but please keep us updated]

N/A.

- All authors must complete a disclosure form prior to acceptance. A link to complete the form has been sent to all coauthors. Please provide the editorial office with updated email addresses if necessary

Email addresses provided.